# ACE2 is the critical in vivo receptor for SARS-CoV-2 in a novel COVID-19 mouse model with TNF- and IFNγ-driven immunopathology

Riem Gawish[1†], Philipp Starkl[1†], Lisabeth Pimenov[1], Anastasiya Hladik[1], Karin Lakovits[1], Felicitas Oberndorfer[2], Shane JF Cronin[3], Anna Ohradanova-Repic[4], Gerald Wirnsberger[5], Benedikt Agerer[6], Lukas Endler[6], Tümay Capraz[7], Jan W Perthold[7], Domagoj Cikes[3], Rubina Koglgruber[3], Astrid Hagelkruys[3], Nuria Montserrat[8,9], Ali Mirazimi[10,11], Louis Boon[12], Hannes Stockinger[4], Andreas Bergthaler[6], Chris Oostenbrink[7], Josef M Penninger[3,13], Sylvia Knapp[1]*

[1]Laboratory of Infection Biology, Department of Medicine I, Medical University of Vienna, Vienna, Austria; [2]Department of Pathology, Medical University of Vienna, Vienna, Austria; [3]Institute of Molecular Biotechnology of the Austrian Academy of Sciences, Vienna, Austria; [4]Molecular Immunology Unit, Institute for Hygiene and Applied Immunology, Center for Pathophysiology, Infectiology and Immunology, Medical University of Vienna, Vienna, Austria; [5]Aperion Biologics, Vienna, Austria; [6]CeMM, Research Center for Molecular Medicine of the Austrian Academy of Sciences, Vienna, Austria; [7]Institute of Molecular Modeling and Simulation, Department of Material Sciences and Process Engineering, University of Natural Resources and Life Sciences, Vienna, Austria; [8]Pluripotency for Organ Regeneration, Institute for Bioengineering of Catalonia (IBEC), The Barcelona Institute of Technology (BIST), Catalan Institution for Research and Advanced Studies (ICREA), Barcelona, Spain; [9]Centro de Investigación Biomédica en Red en Bioingeniería, Biomateriales y Nanomedicina, Madrid, Spain; [10]Karolinska Institute and Karolinska University Hospital, Department of Laboratory Medicine, Unit of Clinical Microbiology, Stockholm, Sweden; [11]National Veterinary Institute, Uppsala, Sweden; [12]Polpharma Biologics, Utrecht, Netherlands; [13]Department of Medical Genetics, Life Sciences Institute, University of British Columbia, Vancouver, Canada

*For correspondence:
Sylvia.knapp@meduniwien.ac.at

†These authors contributed equally to this work

**Abstract** Despite tremendous progress in the understanding of COVID-19, mechanistic insight into immunological, disease-driving factors remains limited. We generated maVie16, a mouse-adapted SARS-CoV-2, by serial passaging of a human isolate. In silico modeling revealed how only three Spike mutations of maVie16 enhanced interaction with murine ACE2. maVie16 induced profound pathology in BALB/c and C57BL/6 mice, and the resulting mouse COVID-19 (mCOVID-19) replicated critical aspects of human disease, including early lymphopenia, pulmonary immune cell infiltration, pneumonia, and specific adaptive immunity. Inhibition of the proinflammatory cytokines IFNγ and TNF substantially reduced immunopathology. Importantly, genetic ACE2-deficiency completely prevented mCOVID-19 development. Finally, inhalation therapy with recombinant ACE2 fully protected mice from mCOVID-19, revealing a novel and efficient treatment. Thus, we here present maVie16 as a new tool to model COVID-19 for the discovery of new therapies and show

that disease severity is determined by cytokine-driven immunopathology and critically dependent on ACE2 in vivo.

## Editor's evaluation

To establish a mouse model for the SARS-CoV-2 infection, Gawish and colleagues performed serial passage of a human virus isolate in mice. They show that the mouse-adapted SARS-CoV-2 variant remains dependent on ACE2 for efficient infection, recapitulates some clinical characteristics of COVID-19, and acquired some changes also found in the Omicron variant of concern. Finally, the authors demonstrate that inhalation of recombinant ACE2 protected mice from mouse COVID-19, suggesting that this model will be useful for the testing of antiviral agents.

## Introduction

SARS-CoV-2 was identified as the causative agent of COVID-19 in December 2019 and has since (until August 5, 2021) infected 200 million people and caused 4.3 million confirmed deaths worldwide (*WHO, 2021b*). Upon adaptation to humans, SARS-CoV-2 evolution gave rise to multiple variants including novel SARS-CoV-2 variants of concern (VOCs), characterized by increased transmissibility and/or virulence, or reduced effectiveness of countermeasures such as vaccinations or antibody therapies (*WHO, 2021a*). Clinical symptoms upon SARS-CoV-2 infection show a wide range, from asymptomatic disease to critical illness (*Chen et al., 2020*; *Paranjpe et al., 2020*). Severe COVID-19 is characterized by progressive respiratory failure, often necessitating hospitalization and mechanical ventilation. Fatal disease is linked to acute respiratory distress syndrome (ARDS), associated with activation of inflammation and thrombosis, often resulting in multiorgan failure (*Chen et al., 2020*; *Paranjpe et al., 2020*). Epidemiologically, several risk factors, such as age, male sex, diabetes, and obesity, have been found to be associated with the development of severe COVID-19 (*Richardson et al., 2020*; *Zhou et al., 2020*). However, early immunological determinants driving disease severity and outcome of SARS-CoV-2 infection remain elusive.

While SARS-CoV-2 replicates primarily in nasopharyngeal and type II alveolar epithelial cells in the lower respiratory tract, recent data point towards a broader cellular tropism under certain conditions (*Sungnak et al., 2020*; *Ziegler, 2020*). This largely correlates with angiotensin-converting enzyme-2 (ACE2) expression, the entry receptor for SARS-CoV-2 (*Hoffmann et al., 2020*), and TMPRSS2 expression, a protease, which supports viral cell entry (*Hoffmann et al., 2020*; *Liu et al., 2021*; *Murgolo et al., 2021*). Severe disease seems to be driven by a circuit of T cells and macrophages causing a cytokine release syndrome, immunothrombosis, and resulting tissue damage that resembles macrophage activation syndrome (*Mangalmurti and Hunter, 2020*). As such, IFNγ released by T cells drives the emergence of pathologically activated, hyperinflammatory monocytes and macrophages that produce large amounts of interleukin (IL)-6, TNFα, and IL-1β (*Grant et al., 2021*). Although numerous studies investigated immune responses upon SARS-CoV-2 infection in humans, these studies are largely correlations in hospitalized patients with moderate to severe disease (*Merad et al., 2021*).

Mechanistic data on early immunological events driving COVID-19 remain sparse due to limited access to human samples and the scarcity of robust small animal models. Notably, mice are resistant to SARS-CoV-2 infection due to phylogenetic differences in ACE2 (*Damas et al., 2020*; *Parolin et al., 2021*). To circumvent the missing sensitivity of conventional mice, some groups studying SARS-CoV-2-related immune mechanisms take advantage of KRT18-hACE2 transgenic mice, which express human ACE2 under the epithelial cytokeratin-18 (KRT18) promoter (*Winkler et al., 2020*). However, KRT18-hACE2 mice are overly susceptible as they develop deadly SARS-CoV-2 encephalitis (in addition to pneumonia) in response to low infection doses due to nonphysiological and broad (over)expression of hACE2 (*Kumari et al., 2021*). Recently, four mouse-adapted SARS-CoV-2 strains have been generated by either genetic engineering (*Dinnon et al., 2020*) and/or serial passaging (*Gu et al., 2020*; *Huang et al., 2021*; *Leist et al., 2020*). However, while infectivity is achieved with all of these strains, only two (*Huang et al., 2021*; *Leist et al., 2020*) cause a pathology that mimics human disease, and pathological mechanisms are still poorly understood. Of note, the VOCs Beta/B.1.351 (*Tegally et al., 2020*) and Gamma/P1 *Voloch, 2020* have

been recently found to productively replicate in mice without causing pneumonia (*Montagutelli et al., 2021*), indicating that additional mutations might further increase mouse infectivity and pathogenicity.

In this study, we report the generation of a novel mouse-adapted SARS-CoV-2 virus, **m**ouse-**a**dapted SARS-CoV-2 virus **Vie**nna, passage **16** (maVie16), which causes a disease reminiscent of COVID-19 in humans in two distinct wild-type mouse strains, BALB/c and C57BL/6 mice. We find that the inflammatory cytokines TNF and IFNγ are important drivers of maVie16 pathogenicity. Moreover, we provide the first genetic in vivo proof that ACE2 is the critical SARS-CoV-2 receptor. Finally, therapeutic inhalation of soluble ACE2 completely prevented disease. Our novel mouse maVie16 infection model is thus a useful and relevant model to study immunology and determinants of SARS-CoV-2 infections and COVID-19 in vivo.

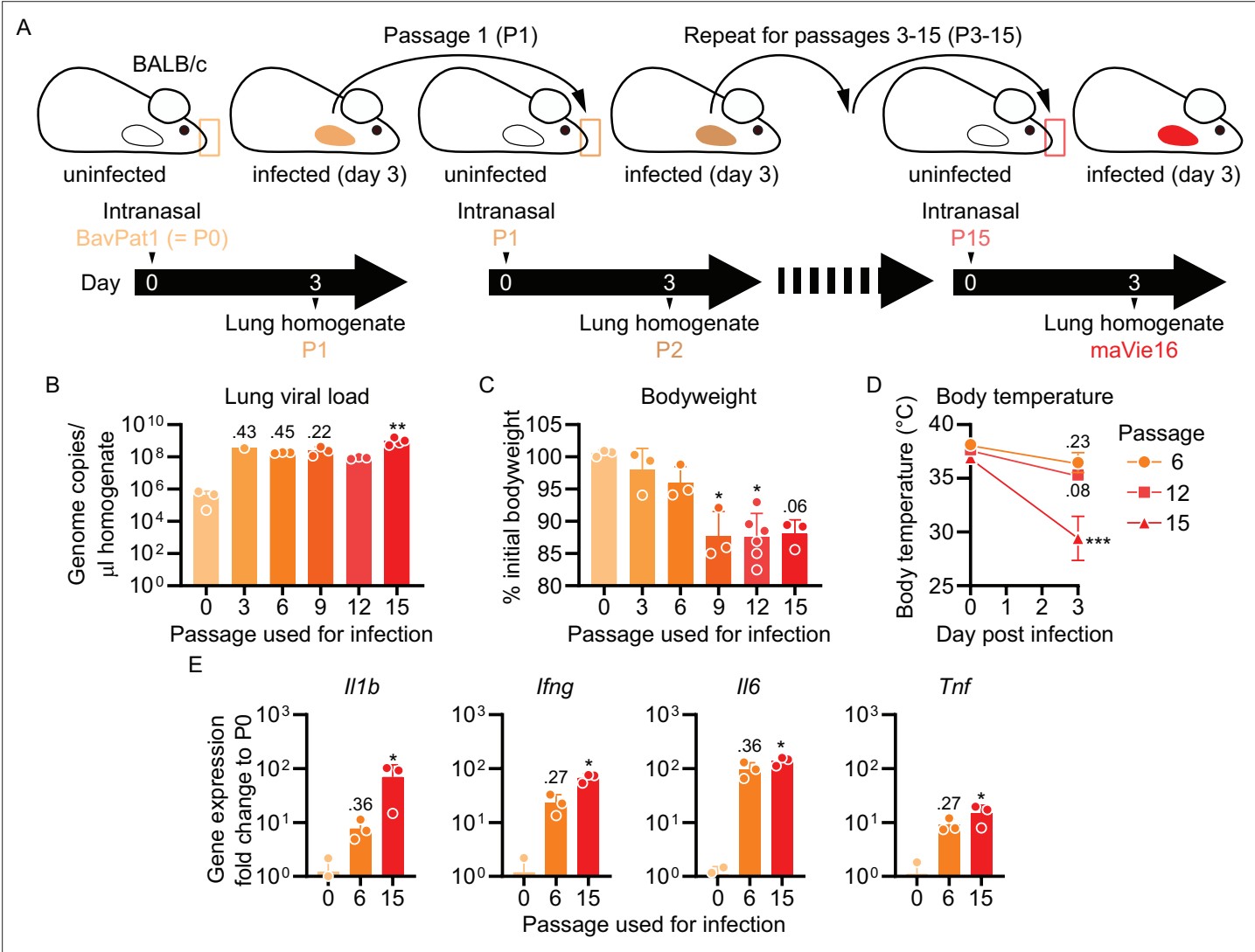

**Figure 1.** Serial pulmonary passaging of SARS-CoV-2 through BALB/c mice leads to mouse adaptation and generation of the mouse-virulent virus maVie16. (**A**) Experimental strategy for generation of maVie16. BALB/c mice were intranasally inoculated with BavPat1 (passage 0/P0), followed by serial passaging of virus-containing cell-free lung homogenates of infected mice every 3 days. Passaging was repeated 15 times. (**B**) Lung tissue virus genome copy numbers (determined by real-time PCR) of mice 3 days after infection with virus of different passages as indicated. (**C**) Body weight (percentage of initial) of mice 3 days after infection. (**D**) Body temperature before and 3 days after infection. (**E**) Lung tissue expression fold change (compared to P0 mean; analyzed by real-time PCR) of indicated genes 3 days after infection. (**B–E**) n = 1–3; (**B, C, E**) symbols represent individual mice; Kruskal–Wallis test (vs. P0) with Dunn's multiple comparisons test; (**D**) mean ± SD; two-way ANOVA with Sidak test (vs. the respective initial body temperature); *p≤0.05; **p≤0.01; ***p≤0.001; numbers above bars show the actual p-value.

## Results

### Generation of the mouse-adapted SARS-CoV-2 strain maVie16

To generate a virus pathogenic to mice, we infected BALB/c mice with a human SARS-CoV-2 isolate (BetaCoV/Munich/BavPat1/2020; from here on referred to as BavPat1; *Rothe et al., 2020*), followed by serial passaging of virus-containing cell-free lung homogenates of infected mice every 3 days (*Figure 1A*). Infectivity was quickly established after the first few passages, as indicated by increasing SARS-CoV-2 genome copies detected in the lungs of infected mice (*Figure 1B*). While the viral load did not further change after passage 3, a progressive loss of body weight of mice infected with later stage passaged SARS-CoV-2 indicated enhanced pathogenicity of the virus (*Figure 1C*). Weight loss is a sign of disease severity in many rodent models of infection and related to anorexia and/or cachexia (*Baazim et al., 2021*). Mice infected with passage 15 furthermore exhibited a severe drop in body temperature (*Figure 1D*), which is the regulatory response of mice to severe inflammation at room temperature (*Garami et al., 2018*). The transcriptional analysis of pulmonary inflammatory genes (including *Il1b*, *Ifng*, *Il6*, and *Tnf*) 3 days after infection with the different passages revealed a progressively increasing inflammatory response (*Figure 1E*). These results indicate that our serial passaging protocol has allowed us to generate a highly infectious and pathogenic mouse-adapted SARS-CoV-2 variant (isolated from lungs of mice infected with passage 15), which we refer to as maVie16.

### Genetic evolution of the maVie16 Spike protein

To elucidate the mechanisms of mouse adaptation of maVie16, we sequenced SARS-CoV-2 genomes at all passages and compared the viral sequences to the original BavPat1 isolate. Despite the substantially increased virulence in mice, we found only a limited number of mutations with high (>0.5) allele frequencies, located in the Spike protein, open reading frame (ORF)1AB, and envelope (*Figure 2A*). Given the importance of Spike for SARS-CoV-2 infectivity, we were particularly interested in the three de novo mutations within its receptor binding domain (RBD) (*Figure 2B*). Most prominently, we observed the immediate appearance of Q498H after passage 1 (*Figure 2B*), which correlated with the early increase in viral load (and hence viral propagation) in infected mice (*Figure 1B*). After passage 12, we detected a glutamine to arginine exchange at position Q493R (*Figure 2B*). Importantly, mutations at Q493 and Q498 have been also reported in other mouse-adapted SARS-CoV-2 strains (*Dinnon et al., 2020*; *Huang et al., 2021*; *Leist et al., 2020*; *Wang et al., 2020b*), suggesting a critical role in Spike adaptation to the murine ACE2 receptor. With the appearance of Omicron/B.1.1.529, the most recent VOC, Q493, and Q498 mutations were detected in a human SARS-CoV-2 variant for the first time. Mutations in K417 (which appeared around passage 10; *Figure 2B*) have also been reported in Omicron/B.1.1.529 and two other human VOCs, that is, Beta/B.1.351 (K417N) (*Tegally et al., 2020*) and Gamma/P1 (K417T) (*Voloch, 2020*).

Structural analyses of mouse and human ACE2 (mACE2 and hACE2, respectively) have shown that the surface of mACE2 consists of more negative charged amino acids than that of hACE2 (*Figure 2—figure supplement 1A*; *Rodrigues et al., 2020*). Indeed, the exchange of two glutamines (Q493 and Q498) to arginine and histidine, respectively, in the maVie16 Spike results in a positive charged surface (*Figure 2C*). Detailed structural modeling of the BavPat1 versus the maVie16 Spike protein interface with mACE2 revealed that the maVie16 Spike Q498H mutation would strengthen its interaction with mACE2 at amino acid D38, otherwise forming an intramolecular salt bridge with mACE2 H353 (*Figure 2D*). Similarly, the newly introduced arginine at maVie16 Spike position 493 is predicted to efficiently interact with the negatively charged mACE2 amino acid E35, thus further stabilizing the maVie16 Spike/mACE2 interaction by neutralizing the otherwise unpaired negative charge of mACE2 glutamic acid (*Figure 2D*). Finally, the threonine at Spike position 417 (T417) of maVie16's is predicted to enhance the interaction with the neutrally charged asparagine at position 30 of mACE2 while the aspartic acid at this position in hACE2 forms a salt bridge with lysine K417 of the original *BavPat1* isolate. Since the observed Spike mutations are distant from putative cleavage sites (*Takeda, 2022*), it is likely that mACE2 affinity is increased without affecting proteolytic processing by proteases such as furin or TMPRSS2 (*Takeda, 2022*). Of note, maVie16 showed similar propagation kinetics in Vero and Caco-2 cells as compared to BavPat1 (*Figure 2—figure supplement 1B*). Thus, while the interaction with mACE2 is enhanced, the infectivity of human cells remained unchanged for maVie16 and both simian and human cells continue to serve as a useful tool for maVie16 propagation and titer determination. Taken together, our data indicate that only three distinct Spike RBD mutations introduced

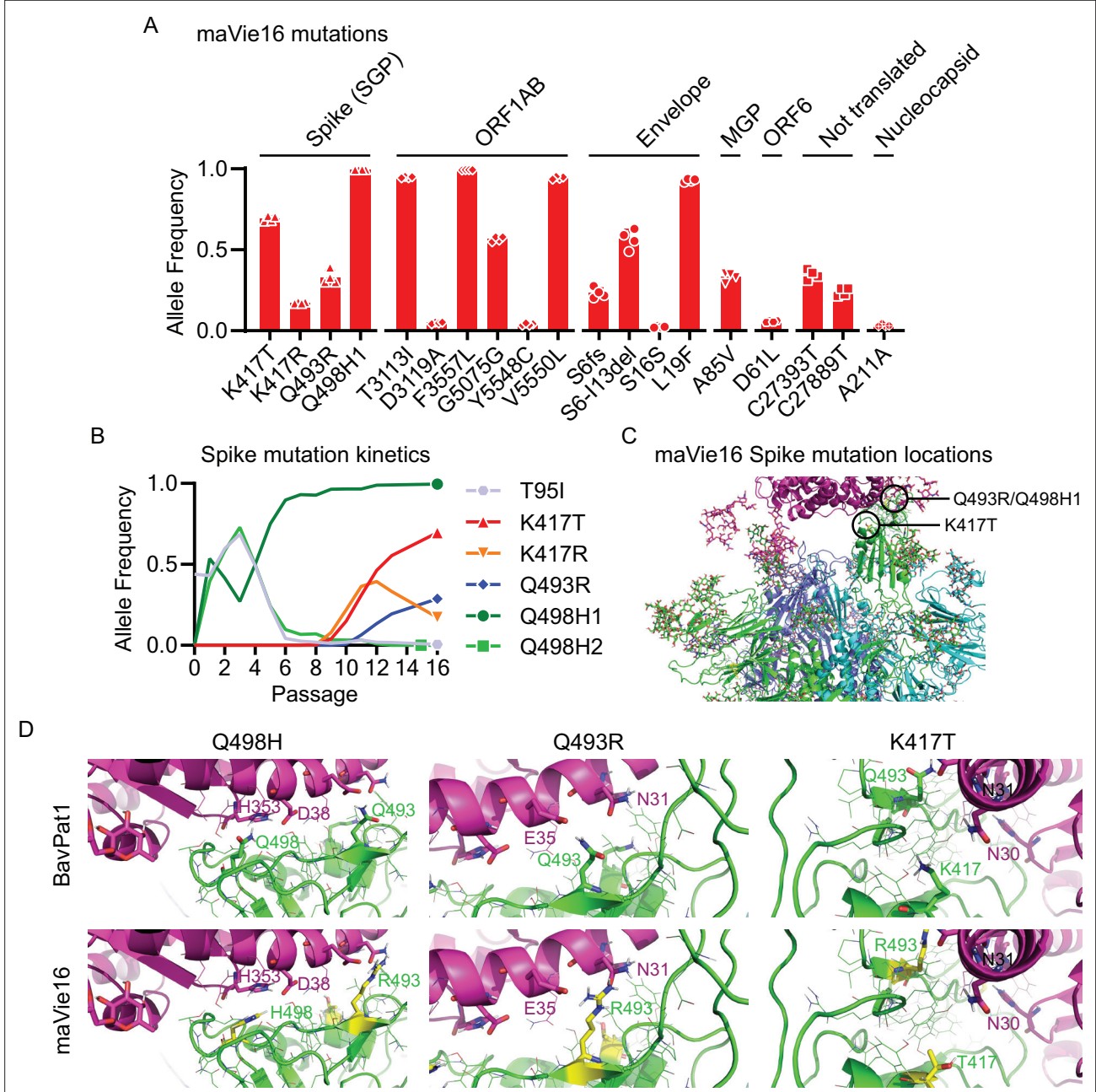

**Figure 2.** maVie16 possesses a distinct pattern of mutations and mediates in vivo pathology via angiotensin-converting enzyme-2 (ACE2). (**A**) Overview of allele frequencies of mutated amino acids detected in maVie16 by sequencing. Labels on top indicate the associated protein (SGP, Spike glycoprotein; ORF, open reading frame; MGP, membrane glycoprotein). (**B**) Spike protein mutation dynamics. (**C**) Modeling and location of Spike mutations. Spike trimer in cyan blue and green, mACE2 in magenta cartoon representation. Glycans in stick representations. (**D**) Modeling of specific *BavPat1* (upper row) and maVie16 (lower row) amino acid regions in green (respective mutated positions are highlighted in yellow and labeled in green) and their interaction with mouse ACE2 (in magenta; positions of interest are labeled in magenta or black).

The online version of this article includes the following figure supplement(s) for figure 2:

**Figure supplement 1.** Mouse versus human angiotensin-converting enzyme-2 (ACE2) glycosylation and maVie16 in vitro proliferation.

during passaging could substantially enhance interaction with mACE2, without obvious effects on the interaction with hACE2.

## maVie16 causes severe pneumonia in BALB/c and C57BL/6 mice

We next performed dose–response experiments of maVie16 infections in BALB/c (B/c) and, as a commonly used mouse strain for genetic engineering, C57BL/6 (B/6) mice and monitored weight and body temperatures over 7 days. In BALB/c mice, maVie16 was highly pathogenic and caused profound weight loss, starting around day 3 post infection (p.i.) at a dose of $4 \times 10^3$ TCID$_{50}$ (*Figure 3A*), whereas all tested higher doses caused an earlier weight loss (day 2 p.i.) and lethality starting at day 4 p.i. (*Figure 3B*). Notably, the loss of body weight correlated with hypothermia (*Figure 3C*). In contrast to BALB/c mice, maVie16 infection did not cause lethality nor significant hypothermia in C57BL/6 animals, albeit $1 \times 10^5$ (and higher) TCID$_{50}$ induced a profound but transient body weight loss of about 15% (*Figure 3D–F*). Interestingly, both mouse strains infected with $1 \times 10^5$ TCID$_{50}$ showed similar body weight loss (approximately 15%) at day 3 p.i., but while disease severity further increased in BALB/c animals, C57BL/6 mice recovered. In line with these data, BALB/c animals showed more severe lung pathologies compared to C57BL/6 mice, as revealed by histological analyses of lungs on day 3 (*Figure 3G and H*). While the cumulative histological score revealed no differences between the strains, diffuse alveolar damage, characterized by alveolar collapse and septal thickening, was more pronounced in BALB/c mice (*Figure 3I*).

To compare our data with a widely used mouse model of COVID-19, we infected KRT18-hACE2 transgenic mice that express human ACE2 under control of the human keratin 18 promoter (*McCray et al., 2007*; *Oladunni et al., 2020*) with either $1 \times 10^3$ or $1 \times 10^4$ TCID$_{50}$ of BavPat1 and monitored the disease course over 7 days. As expected (*Oladunni et al., 2020*), KRT18-hACE2 mice were most sensitive to infection and succumbed to BavPat1 infections (*Figure 3—figure supplement 1A,B*). However, the disease course was profoundly different to that of maVie16-infected wild-type animals. BavPat1-infected KRT18-hACE2 mice appeared healthy until day 4 p.i. and then progressively lost weight until day 7, accompanied by hypothermia and death (*Figure 3—figure supplement 1C–F*). Death in KRT18-hACE2 mice most likely results from severe encephalitis due to expression of hACE2 in neurons where ACE2 is normally not expressed (*Hikmet et al., 2020*). These results show that maVie16 causes severe pathology in BALB/c and profound, but transient, disease in C57BL/6 mice, recapitulating critical aspects (such as transient pneumonia and viral clearance) of human COVID-19.

## Murine anti-SARS-CoV-2 immune responses

To better understand the pathophysiology of mouse COVID-19 (mCOVID-19), we next profiled immune cell dynamics in blood and lungs of C57BL/6 mice during acute infection and upon recovery. maVie16 ($5 \times 10^5$ TCID$_{50}$) infection caused severe blood leukopenia on day 2 p.i. with a prominent reduction of lymphocytes, monocytes, neutrophils, and NK cells (*Figure 4A*, *Figure 4—figure supplement 1*, *Figure 4—figure supplement 2A*). At the same time, we observed an expansion of peripheral plasmacytoid dendritic cells (pDCs) (*Figure 4B*), which are important antiviral effector cells capable of efficient and rapid type I IFN production (*Swiecki and Colonna, 2015*).

Infection led to substantial pulmonary infiltration with leukocytes (*Figure 4C*), paralleling the development of pneumonia. A closer look at the inflammatory cell composition in the lungs revealed a remarkable infiltration with pDCs on day 2 p.i. (*Figure 4C*). Conventional dendritic cells (cDCs) transiently accumulated in the lung on day 5 p.i., followed by T helper and cytotoxic T cell infiltration and recruitment of monocytes on day 7 (*Figure 4C* and *Figure 4—figure supplement 2B*). The abundance of other analyzed immune cell populations did not change substantially, and lung neutrophil numbers dropped around day 2 p.i. to return to baseline levels over the remaining disease course (*Figure 4—figure supplement 2B*). Surprisingly, pulmonary NK cells, which fulfill important antiviral functions (*Björkström et al., 2021*), did not significantly expand in the lungs of maVie16 infected C57BL6 mice (*Figure 4—figure supplement 2B*). In line with an early lung pDC accumulation, we found rapid induction of type I IFN-inducible genes, such as *Eif2ak2* (protein kinase R/PKR) and *ifit1*, accompanied by a profound *Il6* response; the expression of *Tnf*, *Il10*, *Il1b,* and *Tgfb* peaked between day 5 and 7 p.i. (*Figure 4D* and *Figure 4—figure supplement 2C*). The *Ifng* induction reached a maximum on day 7, coinciding with the increased numbers of T cells that are prominent sources of IFNγ (*Figure 4D*).

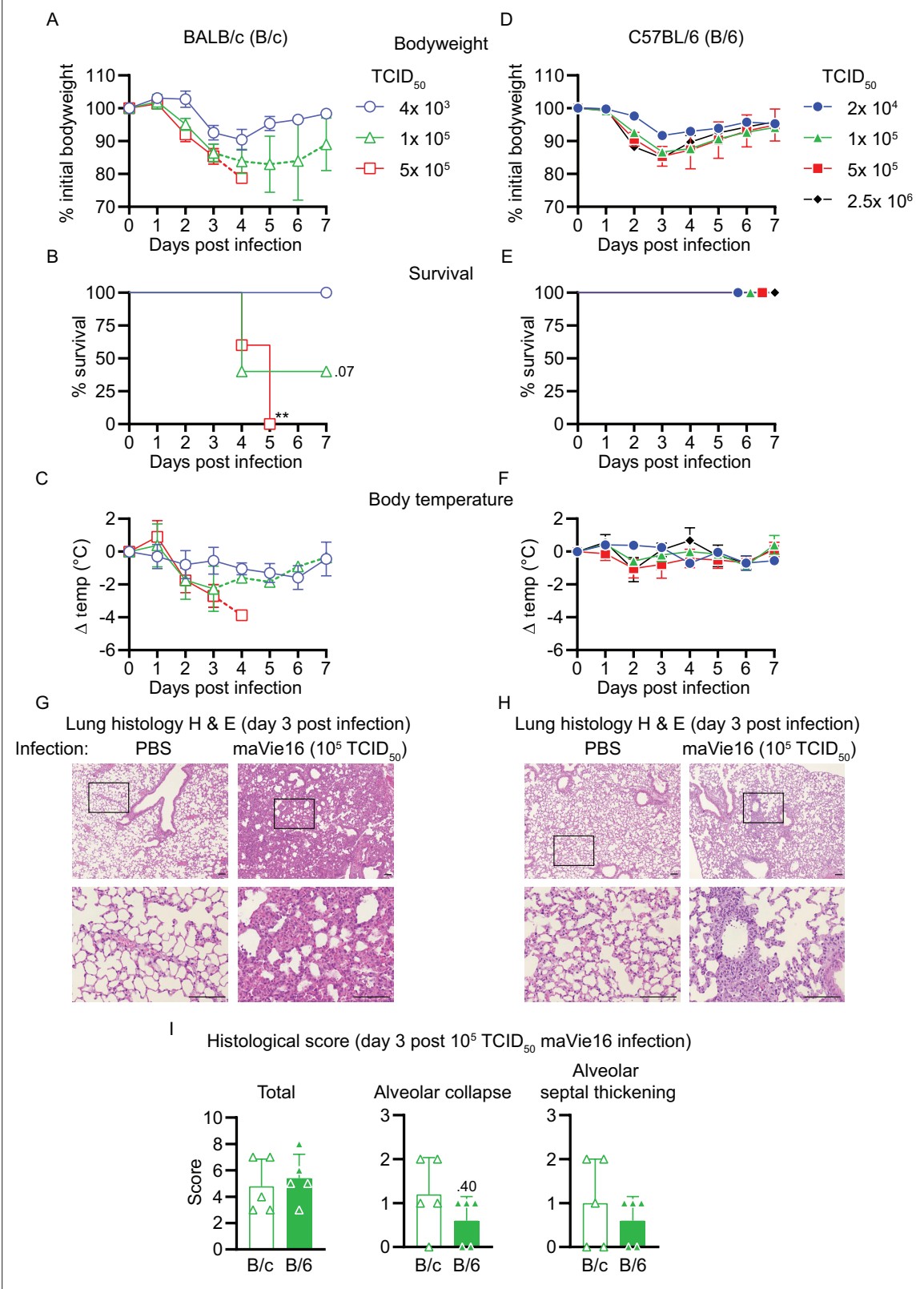

**Figure 3.** Respiratory maVie16 infection causes dose-dependent pathology in BALB/c and C57BL/6 mice. (A–C, G) BALB/c (B/c) or (D–F, H) C57BL/6 (B/6) mice were intranasally inoculated with different doses of maVie16 as indicated and monitored for (A, D) body weight, (B, E) survival and (C, F) body temperature over 7 days; dashed lines in (A) and (C) indicate trajectories of groups lacking full group size due to death of animals (see B). (G, H) Lung sections (hematoxylin and eosin stain) from mice 3 days after infection with $10^5$ TCID$_{50}$ maVie16; black rectangles in the upper pictures indicate the

*Figure 3 continued on next page*

Figure 3 continued

area magnified in the respective lower row picture; scale bars indicate 100 μm. (**I**) Histological score for analysis of lung sections as described in (**G**) and (**H**); symbols represent individual mice; (**A, C, D, F**) mean ± SD; (**B**) Mantel–Cox test (vs. 4 × 10³ TCID$_{50}$); **p≤0.01; the number next to the symbol shows the actual p-value.

The online version of this article includes the following figure supplement(s) for figure 3:

**Figure supplement 1.** Disease kinetics of BavPat1-infected KRT18-hACE2 mice.

In C57BL/6 mice infected with maVie16 (5 × 10⁵ TCID$_{50}$), lung weights significantly increased by day 5–7 p.i. (*Figure 5A*). Lung viral titers peaked at day 2 p.i. and then gradually declined (*Figure 5B*). Immunohistochemical staining for SARS-CoV-2 nucleoprotein on lung slides confirmed the initial presence and subsequent elimination of viral particles (*Figure 5C*). Histological analyses of lung tissue revealed progressive interstitial and perivascular infiltration and signs of diffuse alveolar damage, such as alveolar collapse and septal thickening, peaking from day 5–7, and resolving by day 14 p.i. (*Figure 5C*). Interestingly, vasculitis was observed early after infection (day 2) and resolved at later time points (*Figure 5C and D*). Moreover, most acute inflammatory parameters (including cytokine genes, innate immune cells, and lung tissue weights) had returned close to baseline at day 14 p.i. (*Figures 4 and 5A–D* and *Figure 4—figure supplement 2*), confirming resolution of inflammation. Onset of adaptive immunity, as indicated by expansion of pulmonary cDCs and T helper cells by day 5s and 7 (*Figure 4C*), as well as by increased spleen weight on day 7 p.i. (*Figure 4—figure supplement 2D*), was followed by an efficient humoral immune response, reflected by significantly increased levels of anti-SARS-CoV-2 Spike protein-specific plasma IgG1, IgG2b, and IgA antibodies on day 14 p.i. (*Figure 5E*). These data show that maVie16 infection causes severe, yet transient, lung inflammation in C57BL/6 mice. Furthermore, maVie16 induces a rapid type I IFN response, which correlated with a marked expansion of pDCs.

## Distinct differences in antiviral immunity of BALB/c vs. C57BL/6 mice

In an attempt to elucidate underlying mechanisms for the different susceptibility of BALB/c and C57BL/6 mice, we infected both mouse strains side by side with 1 × 10⁵ TCID$_{50}$ of maVie16 and monitored their respective immune responses 3 days later (as BALB/c and C57BL/6 mice showed comparable weight loss and no mortality at this dose and time point; *Figure 3*). Both mouse strains exhibited a decline in blood leukocytes, including lymphocytes and NK cells, upon infections with maVie16 (*Figure 6A* and *Figure 6—figure supplement 1A*). Also, pulmonary immune cell dynamics did not differ between the strains, except for an expansion of NK cells in BALB/c mice, and elevated pDC numbers in C57BL/6 animals (*Figure 6B* and *Figure 6—figure supplement 1B*). Higher NK cell abundance correlated with increased pulmonary IFNγ levels in BALB/c mice, whereas *Il1b* was slightly higher in C57BL/6 animals 3 days after infection (*Figure 6C* and *Figure 6—figure supplement 1C*). Plasma cytokine levels (*Figure 6—figure supplement 1D*), viral loads (*Figure 6—figure supplement 1E*), as well as spleen and lung weight (*Figure 6—figure supplement 1F*) were comparable between the two mouse strains. These data identify differences in the early local inflammatory response between maVie16-infected BALB/c as compared to C57BL/6 animals.

## A pathogenic role of IFNγ and TNF in disease severity

Several reports illustrated a correlation between systemic cytokine responses, lung injury, and prognosis in humans suffering from severe COVID-19. A particularly detrimental role was attributed to excessive IFNγ levels (*Grant et al., 2021*). Moreover, an earlier study showed that the combined activity of IFNγ and TNF caused inflammatory types of cell death, which was named PANoptosis, that is, pyroptosis, apoptosis, and necroptosis, and that blocking these cytokines improved disease outcomes in KRT18-hACE2 mice infected with human SARS-CoV-2 (*Karki et al., 2021*). In accordance with these reports and our finding of higher IFNγ levels in highly susceptible BALB/c mice, we tested if blocking IFNγ and TNF might reduce disease severity after maVie16 infection. maVie16-infected (1 × 10⁵ TCID$_{50}$) BALB/c mice treated intraperitoneally with a mixture of anti-IFNγ and anti-TNF antibodies (on days 1 and 3 p.i.; *Figure 6D* and *Figure 6—figure supplement 2A*) significantly improved weight loss (*Figure 6E*) and decreased lung weights (*Figure 6G*), the viral load (*Figure 6—figure supplement 2B*), as well as circulating monocyte levels (*Figure 6—figure supplement 2C*) while it had only

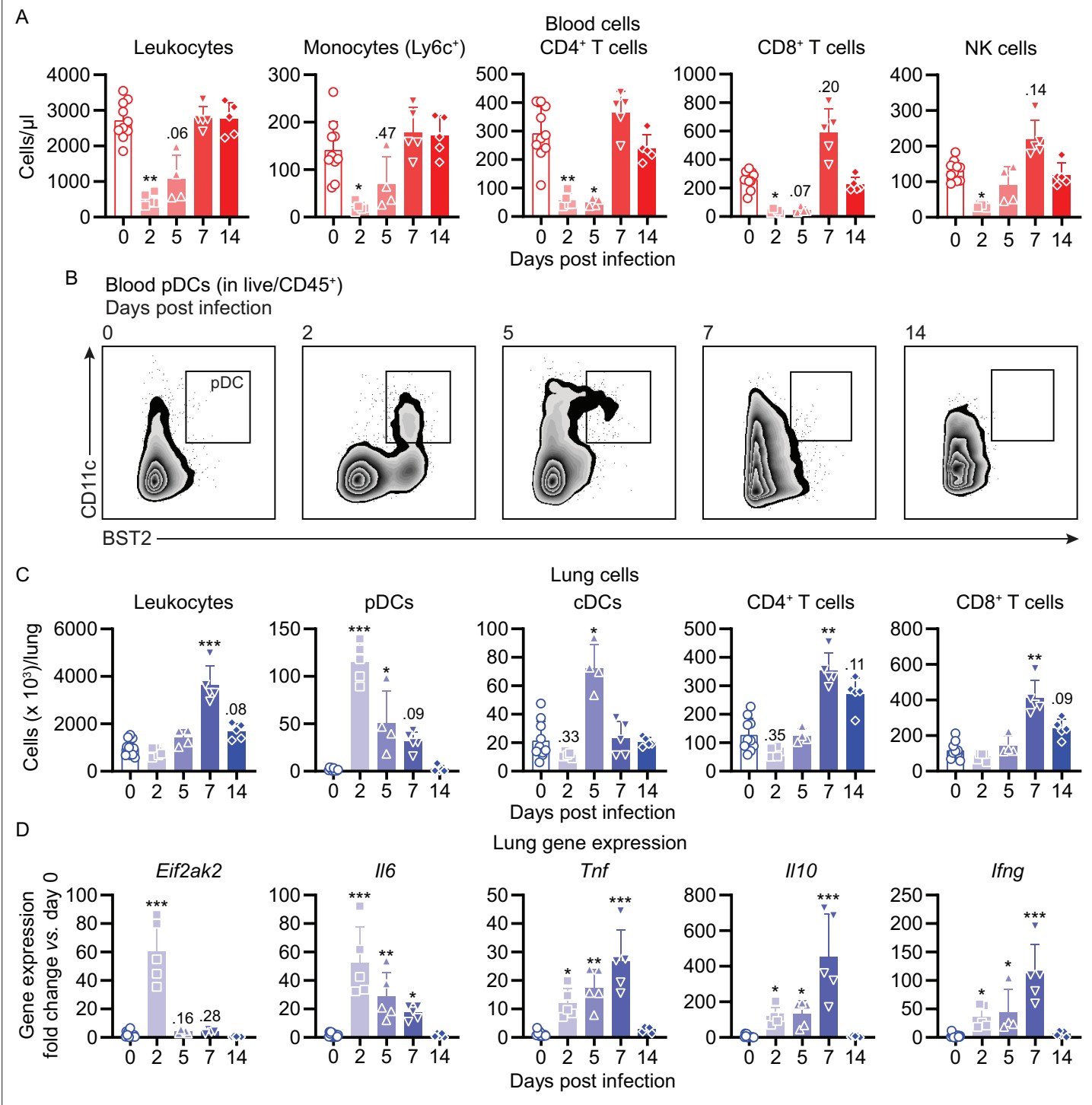

**Figure 4.** Mouse COVID-19 (mCOVID-19) is associated with transient lymphopenia, pulmonary dendritic cell, and T cell infiltration and pneumonia. C57BL/6 mice were intranasally infected with PBS ( = group 0) or $5 \times 10^5$ TCID$_{50}$ maVie16 and sacrificed after 2, 5, 7, or 14 days for subsequent analysis. (**A**) Flow cytometry analysis of blood cell populations. (**B**) Density plot representation of blood plasmacytoid dendritic cells (pDCs; identified as live/CD45$^+$/CD11c$^+$/BST2$^+$) analyzed by flow cytometry. (**C**) Flow cytometry analysis of whole lung cell populations (see *Figure 4—figure supplement 1* for gating strategies). (**D**) Lung tissue expression fold change (compared to group 0 mean; analyzed by real-time PCR) of indicated genes from mice at the respective time points after infection. (**A, C, D**): symbols represent individual mice; Kruskal–Wallis test (vs. group 0) with Dunn's multiple comparisons test; *p≤0.05; **p≤0.01; ***p≤0.001.

The online version of this article includes the following figure supplement(s) for figure 4:

*Figure 4 continued on next page*

*Figure 4 continued*

**Figure supplement 1.** Flow cytometry gating strategy for lung cells.

**Figure supplement 2.** Cellular, transcriptional, and spleen weight kinetics of maVie16-infected C57BL/6 mice.

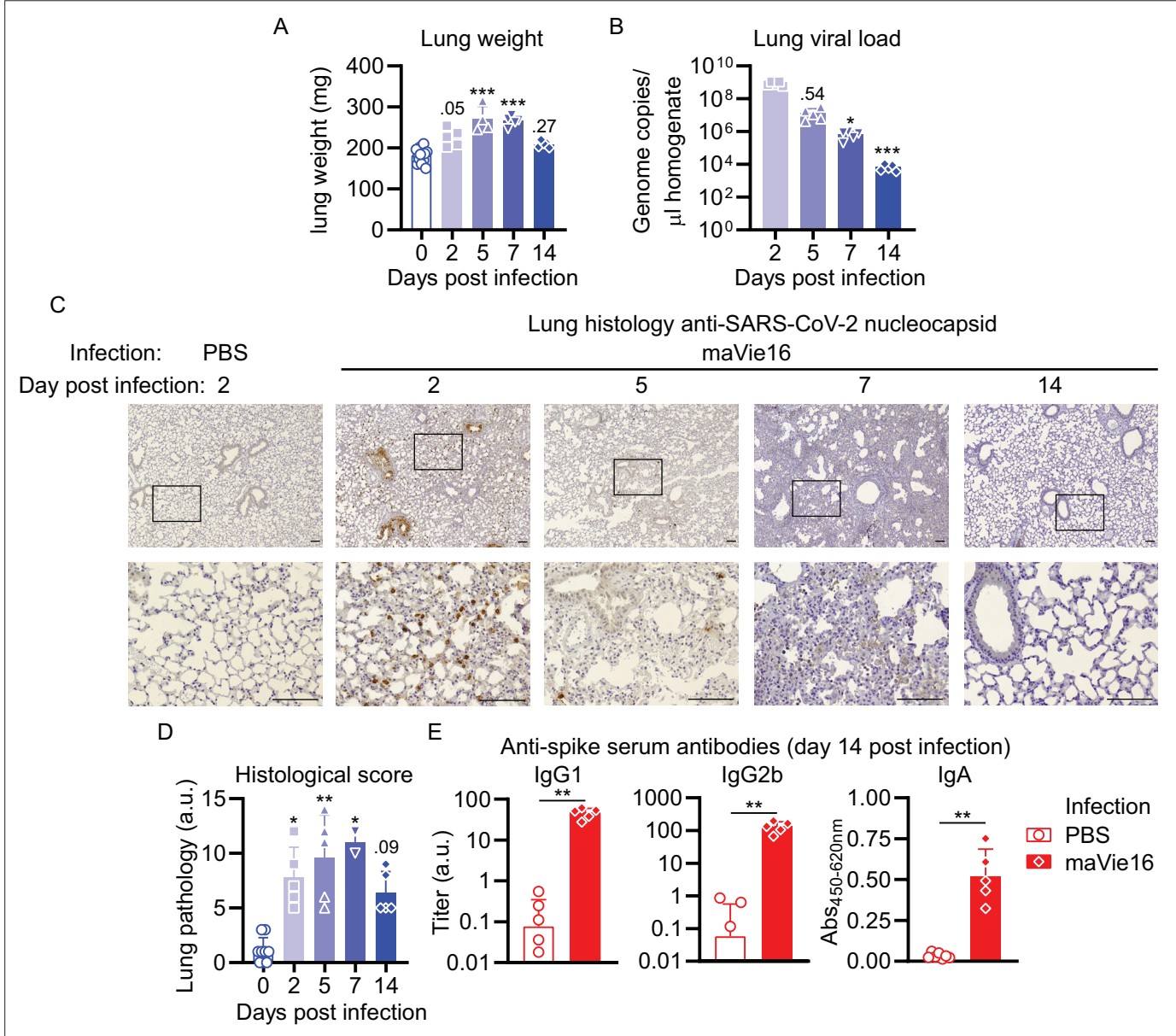

**Figure 5.** Mouse COVID-19 (mCOVID-19) is associated with transient pneumonia and antigen-specific adaptive immunity. C57BL/6 mice were intranasally infected with PBS ( = group 0 or PBS) or $5 \times 10^5$ TCID$_{50}$ maVie16 and sacrificed after 2, 5, 7, or 14 days for subsequent analysis. (**A**) Lung tissue virus genome copy numbers (determined by real-time PCR). (**B**) Lung tissue weight. (**C**) Representative lung immunohistochemistry (anti-SARS-CoV-2 nucleocapsid stain, counterstained with hematoxylin) pictures; black rectangles in the upper pictures indicate the area magnified in the respective lower row picture; scale bars represent 100 µm. (**D**) Lung pathology score based on histological analysis of lung tissue sections. (**E**) Analysis (by ELISA) of SARS-CoV-2 Spike-specific IgG1, IgG2b, and IgA plasma antibody titers 14 days after infection. (**A, B, D, E**) Mean + SD; symbols represent individual mice; (**A, B, D**): Kruskal–Wallis test (vs. [**A**] day 2 or [**B, D**] group 0) with Dunn's multiple comparisons test; (**E**) Mann–Whitney test; *p≤0.05; **p≤0.01; ***p≤0.001.

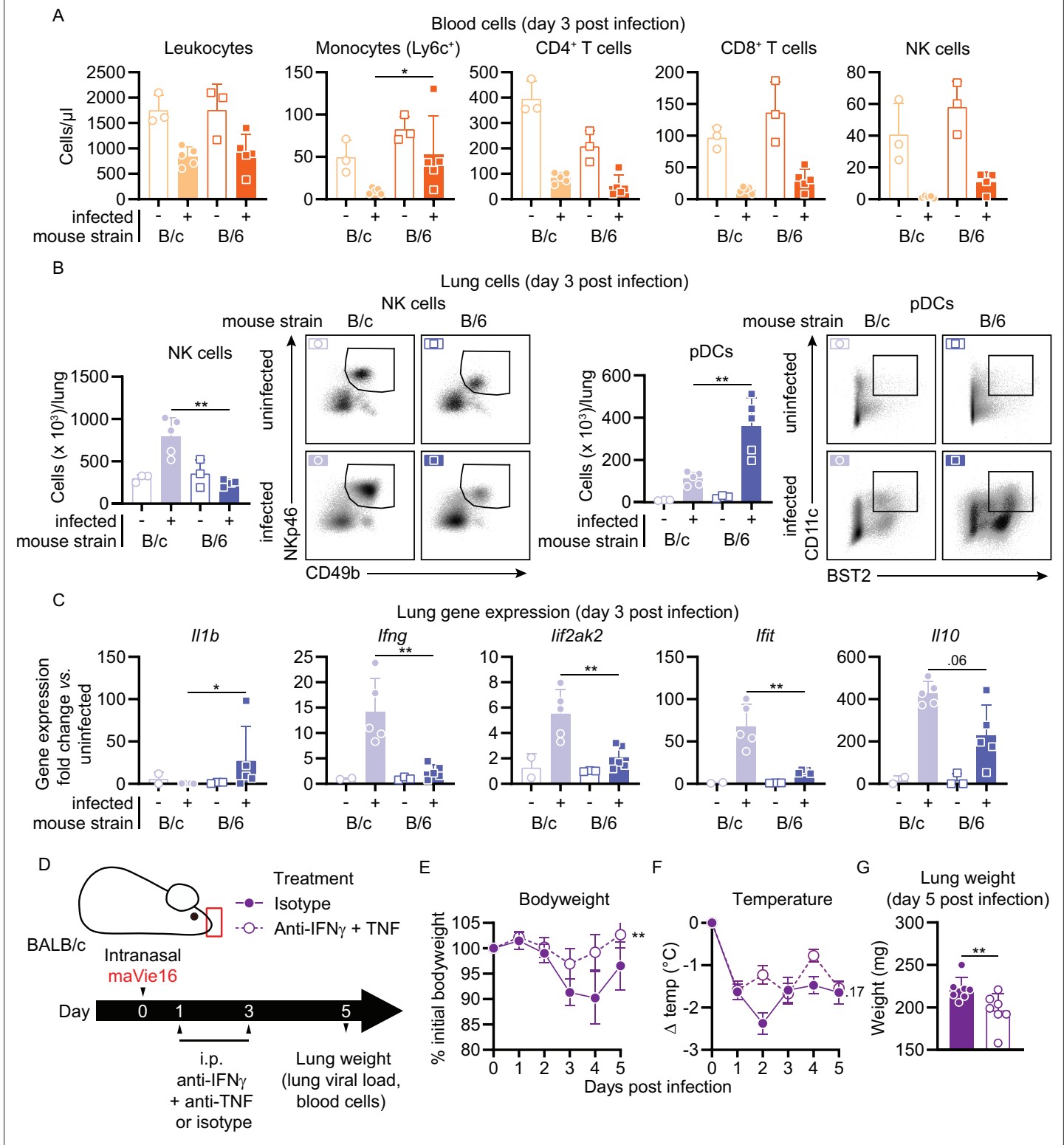

**Figure 6.** BALB/c mouse COVID-19 (mCOVID-19) is associated with an increased NK cell and interferon response and is ameliorated by IFNγ and TNF blockade. BALB/c (B/c) and C57BL/6 (B/6) mice were intranasally inoculated with $10^5$ TCID$_{50}$ maVie16 (+) or PBS (-). Samples for analyses were collected 3 days after infection. (**A**) Flow cytometry analysis of blood cell populations. (**B**) Flow cytometry analysis of whole lung NK cells and plasmacytoid dendritic cells (pDCs). Density plots represent examples of respective cell populations (NK cells pre-gated from live/CD45$^+$/Ly6G$^-$/CD3$^-$; pDCs pre-gated from live/CD45$^+$) (**C**) Lung tissue expression fold change (compared to the respective mean of uninfected samples; analyzed by real-time PCR) of indicated

*Figure 6 continued on next page*

*Figure 6 continued*

genes. (**D**) Experimental scheme for (**E–G**). BALB/c mice were infected with $10^5$ TCID$_{50}$ maVie16 and treated intraperitoneally on days 1 and 3 post infection (p.i.) with a mix of 500 µg anti-IFNγ and anti-TNF or with isotype control antibody. (**E**) Body weight and (**F**) temperature kinetics over 5 days after infection. (**G**) Lung weight on day 5 after infection. (**A–C, G**) Mean + SD; symbols represent individual mice; differences between infected groups were assessed using the Mann–Whitney test; (**E, F**) mean ± SD; two-way ANOVA with Dunnett's multiple comparisons test (vs. the respective initial body weight or temperature); in panels without respective labels, the groups were not significantly different (p<0.05); *p≤0.05; **p≤0.01.

The online version of this article includes the following figure supplement(s) for figure 6:

**Figure supplement 1.** Comparison of cellular, transcriptional, and organ weight of maVie16-infected BALB/c versus C57BL/6 mice.

**Figure supplement 2.** Disease parameters of maVie16-infected Ace2-deficient mice, and of infected BALB/c and C57BL/6 mice treated with anti-IFNγ and -TNF blocking antibodies.

minor effects on body temperature (*Figure 6F*). In C57BL/6 mice, the same treatment had no effects on body or lung weights, or viral load (*Figure 6—figure supplement 2D, E, G and H*), but significantly improved the altered body temperature and circulating leukocyte numbers (*Figure 6—figure supplement 2F and I*). Though other cytokines and chemokines must be involved, these results underline the therapeutic potential of IFNγ and TNF blockade as COVID-19 treatment.

## ACE2 expression is essential for *maVie16* infections

We have previously shown (using *Ace2* mutant mice) that ACE2 is essential for in vivo SARS-CoV infections (*Kuba et al., 2005*). ACE2 is also an important receptor for SARS-CoV-2; however, other receptors have been critically proposed to mediate infections such as neuropilin-1 (*Cantuti-Castelvetri et al., 2020*; *Daly et al., 2020*) or CD147 (*Wang et al., 2020a*, *Wang et al., 2020b*). Having developed the maVie16 infection system, we could therefore ask one of the key questions to understand COVID-19: is ACE2 also the essential entry receptor for SARS-CoV-2 infections in a true in vivo infection model? To answer this question, we infected ACE2-deficient male *Ace2$^{-/y}$* mice (*Figure 7A* and *Figure 7—figure supplement 1A*). In contrast to infected littermate control animals, *Ace2$^{-/y}$* mice were fully protected against infection with $5 \times 10^5$ TCID50 maVie16, maintained stable body weight (*Figure 7B*) and temperature (*Figure 7C*), and were protected from pneumonia development as indicated by lower lung weight (*Figure 7D*) and the absence of any lung pathology (*Figure 7E*). Resistance to infection was further indicated by the complete absence of nucleocapsid-positive cells in lungs of *Ace2$^{-/y}$* animals (*Figure 7E*). In conflict with this observation, we still detected similar viral genome copy numbers in both groups (*Figure 7—figure supplement 1B*), suggesting that this was reflective of initial maVie16 input rather than productive infection. Thus, genetic inactivation of ACE2 protects mice from productive SARS-CoV-2 infections and lung pathologies.

## Inhalation of ACE2 can protect from *maVie16* infections

To test whether ACE2 could also be used therapeutically to protect from maVie16-induced lung pathologies in BALB/c and C57BL/6 mice, we performed prophylactic inhalation treatments with recombinant murine soluble (rms) ACE2. maVie16-infected ($1 \times 10^5$ TCID$_{50}$) BALB/c mice that were treated with rmsACE2 (*Figure 7F* and *Figure 7—figure supplement 1C*) did not lose body weight (*Figure 7G*), developed less hypothermia (*Figure 7H*), exhibited lower lung weights (*Figure 7I*), and were largely protected from pneumonia (day 5 p.i.; *Figure 7J*), had lower lung viral loads (*Figure 7K*), and fewer blood neutrophils (*Figure 7L*) than vehicle-treated animals, indicating marked prevention of mCOVID-19 development. Further, we found fewer SARS-CoV-2 nucleocapsid-positive cells in lungs of BALB/c animals treated with rmsACE2 5 days after infection with maVie16 as compared to controls (*Figure 7—figure supplement 1D*). Similarly, rmsACE2 treatment of maVie16-infected ($5 \times 10^5$ TCID$_{50}$) C57BL/6 mice (*Figure 7—figure supplement 1E*) prevented weight loss and reduced the viral load (*Figure 7—figure supplement 1F and G*, respectively) while it did not affect temperature, lung weight, and blood leukocyte numbers (*Figure 7—figure supplement 1G, H and J*, respectively). Strikingly, inhalative ACE2 therapy was still fully protective when applied 12 hr post infection (*Figure 7M–O*). While mice that received inhalation therapy at later time points post infection still developed disease (*Figure 7N*), mortality was completely prevented or reduced when inhalative treatment was applied 24 hr or 48 hr post infection, respectively (*Figure 7O*). These data show that inhalation of rmsACE2 can markedly prevent maVie16-induced (m)COVID-19.

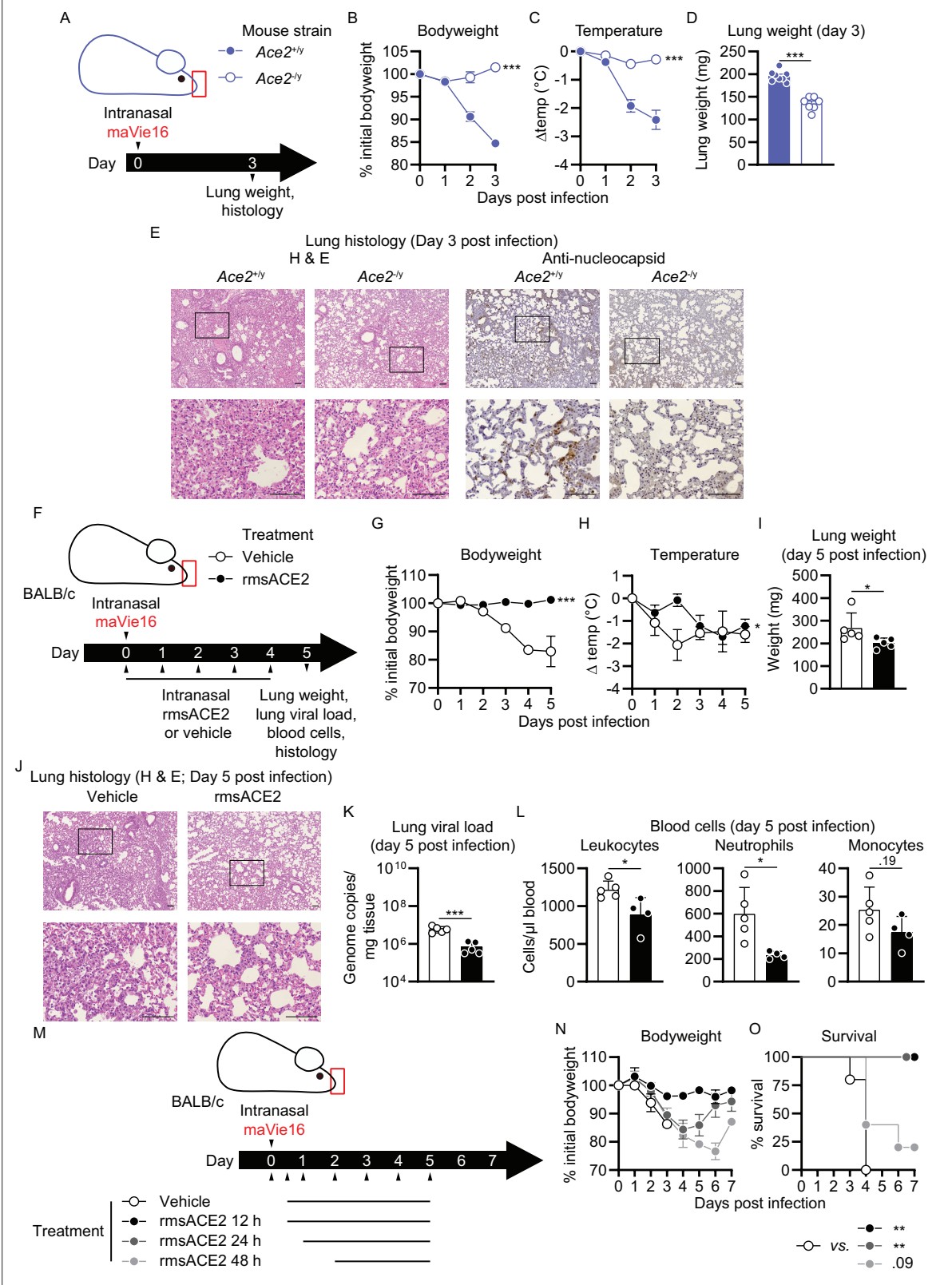

**Figure 7.** Mouse COVID-19 (mCOVID-19) pathology depends on *Ace2* and is improved by recombinant angiotensin-converting enzyme-2 (ACE2) administration. (**A**) Experimental scheme for (**B–E**): male *Ace2*-deficient (*Ace2⁻/ʸ*) or control (*Ace2⁺/ʸ*) mice were infected with $5 \times 10^5$ TCID$_{50}$ maVie16. (**B**) Body weight and (**C**) temperature kinetics over 3 days after infection. (**D**) Lung tissue weight 3 days post infection (p.i.). (**E**) Lung histology 3 days after infection (left panels: hematoxylin and eosin stain; right panels: anti-SARS-CoV-2 nucleocapsid immune-stain); black rectangles in the upper

*Figure 7 continued on next page*

*Figure 7 continued*

pictures indicate the area magnified in the respective lower row picture; scale bars represent 100 µm. (**F**) Experimental scheme for (**G–L**). BALB/c mice were infected with $10^5$ TCID$_{50}$ maVie16 and treated daily intranasally up to day 4 p.i. with 100 µg recombinant murine soluble (rms) ACE2 or vehicle (the first treatment was administered together with virus). (**G**) Body weight and (**H**) temperature kinetics over 5 days after infection. (**I**) Lung weight, (**J**) lung histology (hematoxylin and eosin stain), (**K**) lung viral load, and (**L**) blood cells on day 5 after infection. (**M**) Experimental scheme for (**N**) and (**O**). BALB/c mice were infected with $10^5$ TCID$_{50}$ maVie16 and treated daily, intranasally up to day 5 p.i. with 100 µg rms ACE2 or vehicle. The first treatment was administered either 12 hr, 24 hr, or 48 hr p.i. (**N**) Body weight and (**O**) survival over 7 days of infection. (**B, C, G, H, N, O**) mean ± SD; two-way ANOVA with Dunnett's multiple comparisons test (vs. the respective initial body weight or temperature); (**D, I, K, L**) Mann–Whitney test; * p≤0.05; **p≤0.01; ***p≤0.001; ns, not significant (p>0.05).

The online version of this article includes the following figure supplement(s) for figure 7:

**Figure supplement 1.** Disease parameters of maVie16-infected Ace2-deficient mice, and of infected BALB/c and C57BL/6 mice treated with recombinant mouse angiotensin-converting enzyme-2 (ACE2).

## Discussion

With maVie16, we here generated a novel tool to study and experimentally dissect COVID-19 in vivo. Our passaging protocol led to an initial fast adaptation of the BavPat1 to the new host, indicated by a rapid increase in virus genome copy number found in the lung. This increased mouse infectivity coincided with the appearance of a Spike protein mutation at position 498, which was intentionally introduced using an engineering approach by the Baric group in an earlier attempt to increase mouse specificity of SARS-CoV-2 (*Dinnon et al., 2020*). Interestingly, another maVie16 Spike protein mutation (at position 493) was also found in the mouse-adapted variant of the engineered virus after 10 passages through BALB/c mice (*Gu et al., 2020*; *Huang et al., 2021*; *Leist et al., 2020*) and reported to improve Spike-mAce2 interaction (*Adams et al., 2021*). In support of their apparent mouse specificity, Q498H or Q493R mutations had been only sporadically observed in a few patient samples (16 and 196, respectively, according to the GISAID database; *Shu and McCauley, 2017*) until recently. However, the most recent VOC Omicron/B.1.1.529 harbors the same Q493R mutation and in addition contains Q498R, which leads to the introduction of a basic amino acid (arginine), similar to maVie16's Q498H mutation. The K417T Spike protein mutation was also found in the Gamma/P1 SARS-CoV-2 VOC (*Voloch, 2020*), and the same lysine was changed to an asparagine in the Omicron/B1.1.529 and the Beta/B.1.351 VOC (*Tegally et al., 2020*). Importantly, Beta/B.1.351 and Gamma/P1 SARS-CoV-2 have been shown to replicate in laboratory mice (*Montagutelli et al., 2021*) and in both K417T/N was exclusively observed in conjunction with the mutations N501Y and E484K. While N501Y is suspected to boost infectivity, E484K might contribute to antibody escape and immune evasion. In its interaction with hACE2, E484 is postulated to form a salt bridge with K31 in hACE2, while the neighboring D30 interacts with K417. Mutating K417 and E484 leads to an intramolecular salt bridge between D30 and K31 in hACE2, having only minor effects on receptor interaction (*Cheng et al., 2021*). As outlined above, with aspartic acid replaced by asparagine at position 30 in mACE2, Spike protein amino acid K417 no longer has a strong interaction partner and mutation to T417 in maVie16 is favorable. These data indicate that K417T per se might not itself be involved in immune escape mechanisms, but rather emerged to stabilize the interaction with mACE2.

Our in silico modeling provides strong evidence that these Spike protein mutations in maVie16 support the molecular interaction with mACE2, thereby facilitating infections and the development of severe COVID-19 in mice. At the same time, the multibasic S1/S2 furin cleavage site (681-PRRAR) as well as the S2′ cleavage site (at 815R) were maintained, rendering maVie16 Spike accessible for proteolytic processing by endogenous proteases such as TMPRSS2 or furin, which are essential for productive infection (*Takeda, 2022*) and highly conserved between mice and men (*Vaarala et al., 2001*). Structural analysis did not support an altered binding of maVie16 Spike to hACE2, which was confirmed by propagation experiments with BavPat1 and maVie16 in human Caco-2 cells. In light of the debate about the origin of Omicron/B.1.1.529, we believe that our data, in particular the co-appearance of Q493 and Q498 mutations in the Omicron/B.1.1.529 variant and maVie16, support the idea of Omicron being the result of reverse zoonosis and viral evolution in an animal reservoir (*Kupferschmidt, 2021*).

While inducing profound disease and mortality in BALB/c mice, maVie16 also induced substantial body weight loss and pneumonia in adult C57BL/6 mice. In addition, C57BL/6 mCOVID-19 recapitulated critical immunological aspects observed in human COVID-19, including peripheral

leuko- and lymphopenia, pneumonia, and antigen-specific adaptive immunity. The time to symptom onset was approximately 2–3 days, which is faster than reported in humans (around 5 days) (*Li et al., 2020*) and might be explained by the route of infection and subsequent high viral dose in the lower airways, as opposed to droplet infection of the upper respiratory tract in humans. In addition, mCOVID-19 is characterized by an early pDC mobilization in blood and lung, associated with a fast pulmonary type I IFN response. This correlates with the peak of viral burden, which subsequently declines. However, as others have shown that IFNs play an ambiguous role during COVID-19 (*Israelow et al., 2020*), further experiments (e.g., using IFNAR1-deficient animals or type I IFN blocking antibodies) will be required to address the direct impact of type I IFNs on *maVie16* clearance in vivo. The susceptibility of C57BL/6 animals proves particularly valuable since a large number of genetically modified mice is available on this background. A side-by-side comparison of BALB/c versus C57BL/6 mice revealed increased lung expression of type II IFN and IFN-driven genes in the BALB/c background, supporting the notion that pronounced inflammation is linked to more severe pathology. A detrimental role for cytokines, specifically IFNγ and TNF, had been previously associated with severe COVID-19 and experimentally identified as a risk factor for severe disease in SARS-CoV-2-infected KRT18-hACE2 mice by causing a form of inflammatory cell death termed *PANoptosis* (*Karki et al., 2021*). We also tested this hypothesis and found that administration of IFNγ- and TNF-blocking antibodies significantly reduced disease burden and inflammation in maVie16-infected BALB/c mice. The role of other cytokines and chemokines as well as innate antiviral immunity or defined cell populations can now be genetically and experimentally dissected using our maVie16 infection model.

Several studies based on crystallography (*Wrapp et al., 2020*; *Yan et al., 2020*), modeling (*Rodrigues et al., 2020*), or in vitro experiments (*Cai et al., 2021*; *Hoffmann et al., 2020*; *Monteil et al., 2020*) propose ACE2 as the main entry receptor for SARS-CoV-2. The relevance of ACE2 is further supported by in vivo experiments using KRT18-hACE2 mice (*Oladunni et al., 2020*) or viral vector-mediated hACE2 delivery systems (*Rathnasinghe et al., 2020*). Although these experiments have shown that ACE2 expression is sufficient for infection, definitive proof of an essential in vivo role of ACE2 in COVID-19 has not been provided so far. This is of particular importance because additional receptors have been proposed that could also permit SARS-CoV2 infections (*Cantuti-Castelvetri et al., 2020*; *Daly et al., 2020*; *Wang et al., 2020b*). Using *Ace2*-deficient mice, we here demonstrate the strict dependence of maVie16 infectivity and pathogenicity on ACE2 expression in vivo. Furthermore, we exploited this mechanistic insight and treated animals with rmsACE2 inhalation and observed a substantial protection of BALB/c and C57BL/6 mice from developing mCOVID-19. A recently published genome-wide association study across more than 50,000 COVID-19 patients and more than 700,000 noninfected individuals revealed a rare variant that was associated with downregulated *ACE2* expression and reduced risk of COVID-19 (*Horowitz et al., 2021*), providing additional evidence for the importance of ACE2 for SARS-CoV-2 infection in humans. The principal effectiveness of recombinant human soluble (rhs) ACE2 towards SARS-CoV-2 was demonstrated earlier in organoids as it slowed viral replication in this setting (*Monteil et al., 2020*). It has been recently proposed that soluble ACE2 might enhance SARS-CoV-2 infections (*Yeung et al., 2021*), however, our data do not support this. Moreover, using hamsters (*Higuchi et al., 2021*; *Linsky et al., 2020*) and the KRT18-hACE2 mouse model (*Hassler et al., 2021*), it has also been shown that different forms of ACE2 can protect from disease, providing – together with our novel murine COVID-19 model – preclinical proof of concept that ACE2 can be used as a therapy. RhsACE2, prepared in the same way as rmsACE2 we used for our inhalation experiments (*Monteil et al., 2020*), is now being tested in clinical trials in COVID-19 patients with first promising results (*Zoufaly et al., 2020*). Considering the emergence of variants that can escape in part vaccine efficacy and approved antibody therapies (*Yuan et al., 2021*; *Zhou et al., 2021*) and the fact that these variants have been apparently selected for better ACE2 binding (*Tian et al., 2021*; *Yuan et al., 2021*), ACE2 inhalation has the potential to prevent or treat early stages of SARS-CoV-2 infections irrespective of the virus variant. However, whether inhalable ACE2 indeed constitutes a universal strategy against current and future SARS-CoV-2 variants needs careful testing in clinical trials.

Overall, we here report the development of a novel mouse-adapted SARS-CoV-2, which induces mCOVID-19 in both BALB/c and C57BL/6 backgrounds. Our findings on the essential role of ACE2 in in vivo infections and of recombinant mACE2 administration and IFNγ/TNF blocking as therapeutic

options provide a first glimpse of the potential of this new tool to increase our understanding of COVID-19 in vivo to foster the discovery of novel therapeutic options.

# Materials and methods
## Reagents and resources
**Key resources table**

| Reagent type (species) or resource | Designation | Source or reference | Identifiers | Additional information |
|---|---|---|---|---|
| Strain, strain background (*Mus musculus*, male) | BALB/cJ | Own colony, Jackson Labs | JAX #000651 | |
| Strain, strain background (*M. musculus*, male) | C57BL/6J | Own colony, Jackson Labs | JAX #000664 | |
| Strain, strain background (*M. musculus*, male) | KRT18-hACE2 | Own colony, Jackson Labs | JAX #034860 | |
| Strain, strain background (*M. musculus*, male) | Ace2-/y | Own colony, Jackson Labs *Crackower et al., 2002* | | |
| Strain, strain background (SARS-CoV-2) | BavPat1/2020 | Charité, Berlin, Germany | European Virology Archive # 026V-03883 | |
| Strain, strain background (SARS-CoV-2) | maVie16 | This study | This study | |
| Cell line (*Chlorocebus*) | Vero | ATCC | CCL-81 | |
| Cell line (*Chlorocebus*) | Vero TMPRSS2 (OE) | This study | This study | |
| Cell line (human) | Caco-2 | ATCC | HTB37 | |
| Antibody | Fixable Viability Dye eFluor 780 | Thermo Fisher, eBioscience | Cat# 65-0865-14 | FC (1:3000) |
| Antibody | anti-mouse CD16/32 (monoclonal rat) | BioLegend | Cat# 101320; RRID:AB_1574975 | FC (1:50) |
| Antibody | BV510 anti-mouse CD45 (monoclonal rat) | BioLegend | Cat# 103138; RRID:AB_2563061 | FC (1:200) |
| Antibody | PE anti-mouse CD45 (monoclonal rat) | BioLegend | Cat# 103106; RRID:AB_312971 | FC (1:200) |
| Antibody | PE/dazzle 594 anti-mouse CD45.2 (monoclonal mouse) | BioLegend | Cat# 109845; RRID:AB_2564176 | FC (1:200) |
| Antibody | PE anti-mouse CD45R (monoclonal rat) | BioLegend | Cat# 103208; RRID:AB_312993 | FC (1:200) |
| Antibody | PE anti-mouse F4/80 (monoclonal rat) | BioLegend | Cat# 123110; RRID:AB_893486 | FC (1:200) |

*Continued on next page*

*Continued*

| Reagent type (species) or resource | Designation | Source or reference | Identifiers | Additional information |
|---|---|---|---|---|
| Antibody | Alexa Fluor 700 anti-mouse CD11b (monoclonal rat) | BioLegend | Cat# 101222; RRID:AB_493705 | FC (1:200) |
| Antibody | PE/Cyanine7 anti-mouse CD11b (monoclonal rat) | BioLegend | Cat# 101215; RRID:AB_312798 | FC (1:200) |
| Antibody | Brilliant Violet 605 anti-mouse CD11b (monoclonal rat) | BioLegend | Cat# 101237; RRID:AB_11126744 | FC (1:200) |
| Antibody | PE anti-mouse CD11b (monoclonal rat) | BioLegend | Cat# 101208; RRID:AB_312791 | FC (1:200) |
| Antibody | APC anti-mouse CD11c (monoclonal Armenian hamster) | BioLegend | Cat# 117310; RRID:AB_313779 | FC (1:200) |
| Antibody | PE-Texas Red anti-mouse CD11c (monoclonal Armenian hamster) | Thermo Fisher | Cat# MCD11C17; RRID:AB_10373971 | FC (1:200) |
| Antibody | PE anti-mouse CD11c (monoclonal Armenian hamster) | BD Biosciences | Cat# 557401; RRID:AB_396684 | FC (1:200) |
| Antibody | Brilliant Violet 510 anti-mouse Ly-6C (monoclonal rat) | BioLegend | Cat# 128033; RRID:AB_2562351 | FC (1:200) |
| Antibody | PE/Cy7 anti-mouse Ly-6G (monoclonal rat) | BioLegend | Cat# 127617; RRID:AB_1877262 | FC (1:200) |
| Antibody | FITC anti-mouse Ly-6G (monoclonal rat) | BioLegend | Cat# 127605; RRID:AB_1236488 | FC (1:200) |
| Antibody | BV510 anti-mouse Ly-6G (monoclonal rat) | BioLegend | Cat# 127633; RRID:AB_2562937 | FC (1:200) |
| Antibody | PE anti-mouse Ly6G (monoclonal rat) | BioLegend | Cat# 127608; RRID:AB_1186099 | FC (1:200) |
| Antibody | PerCP-Cy5.5 Siglec-F (monoclonal rat) | BD Biosciences | Cat# 565526; RRID:AB_2739281 | FC (1:100) |
| Antibody | eFluor 450 anti-mouse MHC Class II (I-A/I-E) (monoclonal rat) | Thermo Fisher | Cat# 48-5321-82; RRID:AB_1272204 | FC (1:200) |
| Antibody | BV605 anti-mouse CD115 (monoclonal rat) | BioLegend | Cat# 135517; RRID:AB_2562760 | FC (1:100) |
| Antibody | FITC anti-mouse CD19 (monoclonal rat) | BioLegend | Cat# 115506; RRID:AB_313641 | FC (1:200) |
| Antibody | BV605 anti-mouse CD19 (monoclonal rat) | BioLegend | Cat# 115540; RRID:AB_2563067 | FC (1:200) |
| Antibody | PE anti-mouse CD19 (monoclonal rat) | BioLegend | Cat# 115507; RRID:AB_313642 | FC (1:200) |
| Antibody | FITC anti-mouse CD3 (monoclonal rat) | BioLegend | Cat# 100204; RRID:AB_312661 | FC (1:200) |
| Antibody | PE/Dazzle 594 anti-mouse CD3 (monoclonal rat) | BioLegend | Cat# 100245; RRID:AB_2565882 | FC (1:200) |

*Continued on next page*

*Continued*

| Reagent type (species) or resource | Designation | Source or reference | Identifiers | Additional information |
|---|---|---|---|---|
| Antibody | PerCP/Cy5.5 anti-mouse CD3 (monoclonal rat) | BioLegend | Cat# 100217; RRID:AB_1595597 | FC (1:200) |
| Antibody | eFluor450 anti-mouse CD3 (monoclonal rat) | BioLegend | Cat# 100213; RRID:AB_493644 | FC (1:200) |
| Antibody | PE anti-mouse CD3 (monoclonal rat) | BioLegend | Cat# 100205; RRID:AB_312662 | FC (1:200) |
| Antibody | FITC anti-mouse CD4 (monoclonal rat) | BioLegend | Cat# 100406; RRID:AB_312691 | FC (1:100) |
| Antibody | AF700 anti-mouse CD4 (monoclonal rat) | BioLegend | Cat# 100429; RRID:AB_493698 | FC (1:100) |
| Antibody | PerCP/Cy5.5 anti-mouse CD4 (monoclonal rat) | BioLegend | Cat# 100433; RRID:AB_893330 | FC (1:100) |
| Antibody | PE anti-mouse CD4 (monoclonal rat) | BioLegend | Cat# 100407; RRID:AB_312692 | FC (1:100) |
| Antibody | AF700 anti-mouse CD8a (monoclonal rat) | BioLegend | Cat# 100729; RRID:AB_493702 | FC (1:200) |
| Antibody | Pacific Blue anti-mouse CD8a (monoclonal rat) | BioLegend | Cat# 100728; RRID:AB_493426 | FC (1:200) |
| Antibody | PE anti-mouse CD8a (monoclonal rat) | BioLegend | Cat# 100707; RRID:AB_312746 | FC (1:200) |
| Antibody | PE anti-mouse TCRβ chain (monoclonal Armenian hamster) | BioLegend | Cat# 109207; RRID:AB_313430 | FC (1:200) |
| Antibody | PE anti-mouse TCRγδ (monoclonal Armenian hamster) | BioLegend | Cat# 118107; RRID:AB_313831 | FC (1:200) |
| Antibody | PerCP/Cy5.5 anti-mouse NK-1.1 (monoclonal mouse) | BioLegend | Cat# 108727; RRID:AB_2132706 | FC (1:100) |
| Antibody | APC anti-mouse NK-1.1 (monoclonal mouse) | BioLegend | Cat# 108709; RRID:AB_313396 | FC (1:100) |
| Antibody | PE anti-mouse FcεRIα (monoclonal Armenian hamster) | BioLegend | Cat# 134307; RRID:AB_1626104 | FC (1:100) |
| Antibody | PE/Cy7 anti-mouse CD117 (c-kit) (monoclonal rat) | BioLegend | Cat# 105813; RRID:AB_313222 | FC (1:100) |
| Antibody | BV421 anti-mouse CD193 (CCR3) (monoclonal rat) | BioLegend | Cat# 144517; RRID:AB_2565743 | FC (1:100) |
| Antibody | Alexa Fluor 647 anti-mouse CD49b (monoclonal rat) | BioLegend | Cat# 108912; RRID:AB_492880 | FC (1:100) |
| Antibody | AF700 anti-mouse CD49b (monoclonal rat) | Thermo Fisher | Cat# 56-5971-80; RRID:AB_2574506 | FC (1:100) |
| Antibody | PE/Dazzle 594 anti-mouse CD64 (FCγRI) (monoclonal mouse) | BioLegend | Cat# 139319; RRID:AB_2566558 | FC (1:200) |
| Antibody | AF700 anti-mouse CD43 (monoclonal rat) | BioLegend | Cat# 143213; RRID:AB_2800660 | FC (1:200) |

*Continued on next page*

*Continued*

| Reagent type (species) or resource | Designation | Source or reference | Identifiers | Additional information |
|---|---|---|---|---|
| Antibody | APC anti-mouse CD44 (monoclonal rat) | BioLegend | Cat# 103012; RRID:AB_312963 | FC (1:200) |
| Antibody | BV510 anti-mouse CD138 (monoclonal rat) | BD Biosciences | Cat# 563192; RRID:AB_2738059 | FC (1:100) |
| Antibody | PerCP/Cy5.5 anti-mouse CD21/CD35 (CR2/CR1) (monoclonal rat) | BioLegend | Cat# 123416; RRID:AB_1595490 | FC (1:100) |
| Antibody | AF647 anti-mouse CD23 (monoclonal rat) | BD Biosciences | Cat# 562826; RRID:AB_2737821 | FC (1:200) |
| Antibody | PE-Cy7 anti-mouse CD93 (AA4.1) (monoclonal rat) | Thermo Fisher | Cat# 25-5892-82; RRID:AB_469659 | FC (1:200) |
| Antibody | eFluor450 anti-mouse CD90.2 (Thy1.2) (monoclonal rat) | BioLegend | Cat# 140305; RRID:AB_10645335 | FC (1:200) |
| Antibody | FITC anti-mouse MERTK (monoclonal rat) | BioLegend | Cat# 151504; RRID:AB_2617035 | FC (1:100) |
| Antibody | PE anti-mouse MERTK (monoclonal rat) | BioLegend | Cat# 151505; RRID:AB_2617036 | FC (1:100) |
| Antibody | BV605 anti-mouse CD127 (IL-7α) (monoclonal rat) | BioLegend | Cat# 135025; RRID:AB_2562114 | FC (1:100) |
| Antibody | eFluor450 anti-mouse IgM (monoclonal rat) | Thermo Fisher | Cat# 48-5890-82; RRID:AB_10671539 | FC (1:200) |
| Antibody | FITC anti-mouse IgD (monoclonal rat) | BioLegend | Cat# 405704; RRID:AB_315026 | FC (1:200) |
| Antibody | PE anti-mouse IgA (monoclonal rat) | Thermo Fisher | Cat# 12-4204-83; RRID:AB_465918 | FC (1:200) |
| Antibody | BV605 anti-mouse CD62L (monoclonal rat) | BioLegend | Cat# 104438; RRID:AB_2563058 | FC (1:200) |
| Antibody | APC/Cy7 anti-mouse TER-119 (monoclonal rat) | BioLegend | Cat# 116223; RRID:AB_2137788 | FC (1:200) |
| Antibody | Biotin anti-mouse IgG1 (monoclonal rat) | BD Pharmingen | Cat# 553441 | ELISA (1:1000) |
| Antibody | Biotin anti-mouse IgG2b (monoclonal rat) | BD Pharmingen | Cat# 553393 | ELISA (1:1000) |
| Antibody | Biotin anti-mouse IgA (polyclonal goat) | Southern Biotech | Cat# 1040-08 | ELISA (1:1000) |
| Antibody | Anti-SARS nucleocapsid (polyclonal rabbit) | Novus Biologicals | Cat# NB100-56576 | IHC (1:1000) |
| Antibody | Biotin anti-rabbit IgG (polyclonal goat) | Vector Labs | Cat# BA-1000 | IHC (1:200) |
| Antibody | Anti-mouse TNF (neutralizing) (monoclonal rat) | In house | Clone XT22 | 2 × 500 µg/200 µl/mouse i.p. |
| Antibody | Anti-mouse IFNg (neutralizing) (monoclonal rat) | In house | Clone XMG 1.2 | 2 × 500 µg/200 µl/mouse i.p. |
| Antibody | Anti-ß-galactosidase (IgG1 isotype control) (monoclonal rat) | In house | Clone GL113 | 2 × 500 µg/200 µl/mouse i.p. |

*Continued on next page*

*Continued*

| Reagent type (species) or resource | Designation | Source or reference | Identifiers | Additional information |
|---|---|---|---|---|
| Recombinant DNA reagent | pIRES2-AcGFP1 (Plasmid) | Clontech | Cat# 632435 | |
| Recombinant DNA reagent | pIRES2-SC2quant | This study | | Plasmid containing a PCR-amplified sequence of BavPat1; used as standard for quantification of virus genome copy numbers |
| Recombinant DNA reagent | pBluescript KS(-) | Stratagene | Cat# 212208 | |
| Recombinant DNA reagent | pBMN-I-GFP | Addgene | Plasmid 1736 | |
| Recombinant DNA reagent | pBMN-TMPRSS2-I-GFP | This study | | Plasmid containing the PCR-amplified coding sequence of human TMPRSS2; used to transfect Vero cells to enhance in vitro virus propagation |
| Sequence-based reagent | CoV-F3_XhoI | This study | PCR primers, Microsynth | CTCGAGTTTCCTGGTGATTCTTCTTCAGGT |
| Sequence-based reagent | CoV-R3_BamHI | This study | PCR primers, Microsynth | CCTAGGTCTGAGAGAGGGTCAAGTGC |
| Sequence-based reagent | CoV-F3 | *Gu et al., 2020* | PCR primers, Microsynth | TCCTGGTGATTCTTCTTCAGGT |
| Sequence-based reagent | CoV-R3 | *Gu et al., 2020* | PCR primers, Microsynth | TCTGAGAGAGGGTCAAGTGC |
| Sequence-based reagent | CoV-P3 | *Gu et al., 2020* | PCR primers, Microsynth | AGCTGCAGCACCAGCTGTCCA (FAM/TAMRA-labeled) |
| Sequence-based reagent | Mouse *Adar1*_fwd | This study | PCR primers, Microsynth | GATGACCAGTCTGGAGGTGC |
| Sequence-based reagent | Mouse *Adar1*_rev | This study | PCR primers, Microsynth | GCAGCAAAGCCATGAGATCG |
| Sequence-based reagent | Mouse *Eif2ak2*_fwd | This study | PCR primers, Microsynth | AAGTACAAGCGCTGGCAGAA |
| Sequence-based reagent | Mouse *Eif2ak2*_rev | This study | PCR primers, Microsynth | GCACCGGGTTTTGTATCGAC |
| Sequence-based reagent | Mouse *Ifng*_fwd | This study | PCR primers, Microsynth | ACTGGCAAAAGGATGGTGACA |
| Sequence-based reagent | Mouse *Ifng*_rev | This study | PCR primers, Microsynth | TGGACCTGTGGGTTGTTGAC |
| Sequence-based reagent | Mouse *Ifit1*_fwd | This study | PCR primers, Microsynth | CAGCAACCATGGGAGAGAATGCTGA |
| Sequence-based reagent | Mouse *Ifit1*_rev | This study | PCR primers, Microsynth | GGCACAGTTGCCCCAGGTCG |
| Sequence-based reagent | Mouse *Il1b*_fwd | This study | PCR primers, Microsynth | CAAAATACCTGTGGCCTTGG |
| Sequence-based reagent | Mouse *Il1b*_rev | This study | PCR primers, Microsynth | TACCAGTTGGGGAACTCTGC |
| Sequence-based reagent | Mouse *Il6*_fwd | This study | PCR primers, Microsynth | CCACGGCCTTCCCTACTTCA |
| Sequence-based reagent | Mouse *Il6*_rev | This study | PCR primers, Microsynth | TGCAAGTGCATCGTTGTTC |

*Continued*

| Reagent type (species) or resource | Designation | Source or reference | Identifiers | Additional information |
|---|---|---|---|---|
| Sequence-based reagent | Mouse *Il10*_fwd | This study | PCR primers, Microsynth | TGAGGCGCTGTCATCGATTT |
| Sequence-based reagent | Mouse *Il10*_rev | This study | PCR primers, Microsynth | CATGGCCTTGTAGACACCTT |
| Sequence-based reagent | Mouse *Tgfb*_fwd | This study | PCR primers, Microsynth | AGCCCGAAGCGGACTAT |
| Sequence-based reagent | Mouse *Tgfb*_rev | This study | PCR primers, Microsynth | TCCACATGTTGCTCCACACT |
| Sequence-based reagent | Mouse *Tnf*_fwd | This study | PCR primers, Microsynth | GCGTGGAGCTGAGAGATAACC |
| Sequence-based reagent | Mouse *Tnf*_rev | This study | PCR primers, Microsynth | GATCCCAAAGTAGACCTGCCC |
| Sequence-based reagent | Human TMPRSS2_fwd | This study | PCR primers | AACCTGGGCGCCTGGGA |
| Sequence-based reagent | Human TMPRSS2_rev | This study | PCR primers | ACGTCAAGGACGAAGACCATGTG |
| Peptide, recombinant protein | Collagenase I | Gibco/Thermo Fisher Scientific | 17018029 | |
| Peptide, recombinant protein | DNAse I | Sigma | DN25 | |
| Peptide, recombinant protein | Recombinant SARS-CoV-2 Spike protein ectodomain | Reingard Grabherr, BOKU Vienna *Klausberger et al., 2021* | | |
| Peptide, recombinant protein | mrsACE2 | In house (APEIRON Biologics) *Monteil et al., 2020* | | |
| Commercial assay or kit | ROTI Prep RNA Mini | Carl Roth | Cat# 8485.1 | |
| Commercial assay or kit | QIAamp Viral RNA Mini Kit | QIAGEN | Cat# 52904 | |
| Commercial assay or kit | E.Z.N.A Viral RNA kit | Omega Bio-tek | Cat# R6874 | |
| Commercial assay or kit | qScript cDNA Synthesis Kit | Quantabio | Cat# 95047-500 | |
| Commercial assay or kit | PerfeCTa SYBR Green SuperMix | Quantabio | Cat# 95055 | |
| Commercial assay or kit | Vectastain ABC kit | Vector Labs | Cat# PK-6100 | |
| Commercial assay or kit | DAB Substrate kit | Vector Labs | Cat# SK-4100 | |
| Commercial assay or kit | LEGENDplex Macrophage/ Microglia Panel | BioLegend | Cat# 740846 | |
| Commercial assay or kit | LEGENDplex MU Th Cytokine Panel V02 | BioLegend | Cat# 740741 | |

*Continued on next page*

*Continued*

| Reagent type (species) or resource | Designation | Source or reference | Identifiers | Additional information |
|---|---|---|---|---|
| Chemical compound, drug | Sodium pyruvate (100 mM) | Thermo Fisher Scientific | Cat# 11360070 | |
| Chemical compound, drug | Penicillin streptomycin (10,000 U/ml) | Thermo Fisher Scientific | Cat# 15140122 | |
| Chemical compound, drug | MEM nonessential amino acids (100×) | Thermo Fisher Scientific | Cat# 11140050 | |
| Chemical compound, drug | RLT Plus | QIAGEN | Cat# 1053393 | |
| Chemical compound, drug | 2-Mercaptoethanol | Sigma-Aldrich | Cat# M3148 | |
| Chemical compound, drug | Antigen Unmasking Solution | Vector Labs | Cat# H3300-250 | |
| Chemical compound, drug | Hematoxylin solution (Mayer's) | Sigma-Aldrich | Cat# MHS16 | |
| Software, algorithm | GraphPad Prism 9.1 | GraphPad Software, Inc. | https://www.graphpad.com | |
| Software, algorithm | FlowJo | Becton, Dickinson and Company | https://www.flowjo.com/ | |
| Other | Dulbecco's Modified Eagle's Medium | Thermo Fisher Scientific | Cat# 10564011 | High glucose, GlutaMAX, HEPES |
| Other | Fetal bovine serum | Sigma | Cat# F9665 | |
| Other | Goat serum | Novus Biologicals | Cat# NBP2-23475 | |

## Methods

### Animal studies

For passaging, dose–response and time-course experiments, male 10–12-week-old BALB/cJ and C57BL/6J mice obtained from Janvier or bred and maintained at the animal facility of the Medical University of Vienna were used. Mice expressing human ACE2 under control of the human *KERATIN-18* (KRT18) promoter (Tg[KRT18-Ace2])2[Prlmn] (*McCray et al., 2007*) (herein referred to as KRT18-huACE2 mice) were obtained from Jackson Labs and bred in the local animal facility. Ace-2-deficient (*Ace2*[-/-]) mice were generated as described (*Crackower et al., 2002*) and bred at the animal facility of the Institute of Molecular Biotechnology Austria, Vienna. All experiments involving SARS-CoV-2 or its derivatives were performed in Biosafety Level 3 (BSL-3) facilities at the Medical University of Vienna and performed after approval by the institutional review board of the Austrian Ministry of Sciences (BMBWF-2020-0.253.770) and in accordance with the directives of the EU.

### Cell lines

Simian kidney Vero cells were provided by Christoph Steininger (Medical University of Vienna, ATCC CCL81). To improve in vitro SARS-CoV-2 propagation (*Matsuyama et al., 2020*), Vero cells were transduced as previously described (*Machacek et al., 2016*) with the retroviral expression vector pBMN-I-GFP carrying human TMPRSS2, amplified from the human lung adenocarcinoma cell line Calu-3 (ATCC HTB-55, provided by Walter Berger, Medical University of Vienna) and subcloned via the pBluescript KS(-) plasmid. The transduced cell population was subsequently sorted based on GFP expression to >99% purity, creating Vero[TMPRSS2] cells. Both cell lines were cultured in Dulbecco's Modified Eagle's medium (DMEM, Gibco/Thermo Fisher) supplemented with GlutaMAX, 10% fetal calf serum (FCS; Sigma), 1% MEM nonessential amino acids solution (Thermo Fisher), and 1% sodium pyruvate (Thermo Fisher) and grown at 37°C at 5% $CO_2$.

## SARS-CoV-2 strains, virus propagation, and growth curves

The human SARS-CoV-2 isolate BetaCoV/Munich/BavPat1/2020 (referred to as BavPat1) was kindly provided by Christian Drosten, Charité, Berlin (*Rothe et al., 2020*), and distributed by the European Virology Archive (Ref-SKU: 026V-03883). maVie16 was generated in this study as described. To generate high titer viral stocks, TMPRSS2-overexpressing Vero cells (Vero$^{TMPRSS2}$) were grown on 175 cm$^2$ tissue culture flasks and were infected with BavPat1 or maVie16 at a multiplicity of infection (MOI) of 0.1 in Vero cell culture medium (described above) with reduced (2%) FCS content. The inoculum was calculated based on stock concentrations determined using TCID$_{50}$ assays (described below) assuming that 1 * PFU/ml = 0.7 * TCID$_{50}$/ml applying the Poisson distribution for the estimation of PFU. Supernatants from infected cells were collected after 48–72 hr, clarified by centrifugation (3820 × $g$, 15 min), and stored in aliquots at –80°C. To determine stock concentrations, TCID$_{50}$ assays were conducted by infecting Vero cells with 150 µl of 10-fold serial dilutions of viral samples (cell supernatant, cell-free lung homogenate), followed by incubation at 37°C. Cytopatic effects were analyzed by microscopy after 72 hr and titers were calculated by the method of *Reed and Muench, 1938*. To compare infectivity and viral replication, Vero cells or Caco-2 cells were infected with BavPat1 or maVie16 at a MOI of 0.05 and incubated at 37°C. After 1 hr, the viral inoculum was removed and replaced with medium. At designated time points, supernatants were removed, cleared by centrifugation (500 × $g$, 5 min), and stored at –80°C until further processing for RNA isolation and quantification of genome copies.

## Generation of a mouse-adapted SARS-CoV-2 virus, maVie16

Mouse adaptation of SARS-CoV-2 and generation of maVie16 were achieved by serial passaging through lungs of BALB/c mice (16 passages). Briefly, three anesthetized (isoflurane), male 10–12-week-old BALB/c mice were infected intranasally (i.n.) with 2 × 10$^6$ TCID$_{50}$ BavPat1 in 50 µl. After 3 days, the mice were euthanized and their lungs were homogenized using a rotor/stator homogenizer in eight-fold volume of DMEM. After removal of aliquots for RNA isolation (see below), the lung homogenates were cleared by centrifugation (3820 × $g$, 6 min), filtered using round-bottom tubes with cell-strainer cap (Falcon), pooled, and administered i.n. to the next group of anesthetized naïve male BALB/c mice. This process of i.n. infection and harvest was repeated 15 times, and weight loss and body temperature (measured with a rectal rodent thermometer; Braintree Scientific) were monitored as readouts of viral pathogenicity. Viral load was quantified by detection of SARS-CoV-2 RNA in the cleared lung homogenate described below. The maVie16 stock (collected from the lungs of mice infected with passage 15) was prepared in Vero cells as described above. maVie16 virus will be shared with research labs fulfilling respective safety requirements upon request and execution of a material transfer agreement.

## Mouse model of COVID-19 (mCOVID-19) and tissue collection

For dose–response and time-course experiments, male 10–12-week-old BALB/c mice and C57BL/6 mice were infected i.n. with 50 µl DMEM containing indicated doses of maVie16. Clinical signs of disease (loss of body weight and/or temperature) were monitored at indicated time points. Mice were euthanized by cervical dislocation at indicated endpoints for sample collection or whenever they approached 75% of their starting body weight, in accordance with ethical guidelines.

Blood was collected from euthanized animals from the vena cava and transferred to MiniCollect tubes (Greiner) containing K$_3$EDTA. 75 µl of blood per flow cytometry panel were removed, treated for erythrocyte lysis using ACK buffer (150 mM NH$_4$Cl, 10 mM KHCO$_3$, 0.1 mM Na$_2$EDTA, pH 7.2–7.4), and processed for flow cytometry as described below. The remaining blood was centrifuged 10 min at 1700 × $g$, followed by plasma collection and storage at –80°C.

Whole lungs were harvested and weights recorded. The left lung was assigned to histology and fixed in 7.5% buffered formaline. At least 50 µg of the right lung were kept for tissue homogenization as described above. For total lung RNA, 80 µl of homogenized tissue was lysed in RLT Buffer (QIAGEN) containing 2-mercaptoethanol (Sigma-Aldrich) and stored at –80°C until further use. The remaining homogenized lung tissue was centrifuged (3820 × $g$, 6 min) and the cleared homogenate was frozen at –80°C for subsequent viral RNA isolation (see below). To reprepare lung single-cell suspensions for flow cytometry, the remaining right lungs were minced in gentleMACS tubes with the gentleMACS dissociator (Miltenyi Biotec; program m_lung_01) and digested in 2.5 ml of digestion medium (RPMI medium [Gibco] containing 5% FCS, 250 U/ml collagenase I [Gibco], and 20 U/ml

DNase I [Sigma]) by shaking at 100 rpm for 30 min at 37°C. Digested samples were then homogenized with the gentleMACS dissociator (program m_lung_02) and filtered over a 70 µm cell strainer (BD Biosciences). After centrifugation (500 × $g$, 5 min, 4°C), the cell pellet was treated for 5 min with ACK to lyse erythrocytes. After stopping erythrocyte lysis by addition of PBS 1% bovine serum albumin (BSA; Sigma), cells were passaged over a 40 µm cell strainer (BD Biosciences) and resuspended for subsequent antibody staining (see below).

## RNA isolation and gene expression analysis by real-time PCR

Total RNA was extracted from mouse lung homogenates, and viral RNA was extracted from cleared mouse lung homogenates and cell culture supernatants using the following commercially available kits according to the manufacturer's instructions: for viral RNA extraction, QIAamp Viral RNA Mini Kit (QIAGEN) and E.Z.N.A Viral RNA kit (Omega Bio-tek) were used. Quantification of viral RNA was performed by quantitative reverse transcription PCR (RT-qPCR) as described below. For total RNA extraction, the samples were passed through a QIAshredder (QIAGEN) to reduce viscosity and subsequently RNA was isolated using the ROTI Prep RNA Mini kit (Carl Roth). After isolation, RNA concentration was measured using a NanoVue Plus Spectrophotometer (GE Healthcare). For cDNA synthesis, 0.5 µg RNA from each sample was reverse-transcribed using the qScript cDNA Synthesis Kit (Quantabio). For subsequent qPCR reactions, PerfeCTa SYBR Green SuperMix (Quantabio) was used in a total reaction volume of 15 µl. TaqMan primer/probes for mouse inflammatory genes were purchased from Microsynth and are listed below. RT-PCR was performed on a StepOnePlus Real-Time PCR System (Applied Biosystems) with the following conditions: 95°C for 3 min, followed by 45 amplification cycles (15 s at 95°C) and elongation (1 min at 60°C).

## SARS-CoV-2 quantification by real-time PCR

To generate a standard curve for quantification of virus genome copy numbers, a 144 bp fragment of the SARS-CoV-2 genome was amplified from BavPat1 cDNA (generated using the qScript XLT cDNA supermix [Quanta Bio] with RNA purified with the QIAamp Viral RNA Mini Kit) with simultaneous insertion of XhoI (5' end) and BamHI (3' end) restriction site overhangs by PCR (Touchdown from 70°C to 55°C over 35 cycles) using primers CoV-F3_XhoI and CoV-R3_BamHI and Q5 High-Fidelity DNA Polymerase (New England Biolabs). The purified PCR product was then cloned into pIRES2-GFP1 (Clontech) via directional restriction digest and ligation (XhoI and BamHI Fast Digest enzymes from Thermo Fisher; Ligase from New England Biolabs) to generate the plasmid pIRES2-SC2quant. pIRES2-SC2quant was transfected into One Shot TOP10 Chemically Competent *Escherichia coli*, followed by amplification from single-cell colonies, isolation of plasmid DNA (using QIAPrep Mini and Midi Prep Kits [QIAGEN]) and validation of the construct by sequencing (Microsynth). Based on the molecular weight of the pIRES2-SC2quant plasmid (3344181.19 Da), a $\log^{10}$ dilution series of the plasmid was prepared to contain 0–$10^{10}$ plasmid copies. These dilutions were included on each plate for quantification of viral load in tissue samples by qPCR. The qPCR was performed on a StepOnePlus Real-Time PCR System (Applied Biosystems) using primers CoV-F3 and CoV-R3 with (FAM/TAMRA-labeled) probe CoV-P3 (all at 10 µM and all obtained from Microsynth) and the Ultraplex 1-Step ToughMix ROX (Quantabio), with an initial 10 min 50°C incubation for cDNA synthesis, followed by 40 amplification cycles of denaturation (15 s at 95°C) and elongation (1 min at 60°C) (*Gu et al., 2020*).

## SARS-CoV-2 sequencing

For SARS-CoV-2 genome sequencing, viral RNA was processed as described previously (*Agerer et al., 2021*; *Popa et al., 2020*). Briefly, viral RNA was reverse-transcribed with Superscript IV reverse transcriptase (Thermo Fisher Scientific) and viral sequences were amplified with modified primer pools (*Itokawa et al., 2020*). PCR reactions were pooled and processed for high-throughput sequencing. Amplicons were cleaned up with AMPure XP beads (Beckman Coulter) with a 1:1 ratio. Amplicon concentrations and size distribution were assessed with the Qubit Fluorometric Quantitation system (Life Technologies), and the 2100 Bioanalyzer system (Agilent). Amplicon concentrations were normalized, and sequencing libraries were prepared using the NEBNext Ultra II DNA Library Prep Kit for Illumina (New England Biolabs) according to the manufacturer's instructions. Library concentrations and size distribution were again assessed as indicated previously and pooled into equimolar amounts

for sequencing. Sequencing was carried out on the NovaSeq 6000 platform (Illumina) on a SP flow cell with a read length of 2 × 250 bp in paired-end mode at the Biomedical Sequencing Facility (BSF) of the Medical University of Vienna and CeMM (Research Center for Molecular Medicine of the Austrian Academy of Sciences). Following demultiplexing, FASTQ files were quality controlled using FastQC (v.0.11.8). Trimming of adapter sequences was performed with BBDUK from the BBtools suite (http://jgi.doe.gov/data-and-tools/bbtools). Overlapping read sequences within a pair were corrected for using BBMERGE function from BBTools. Read pairs were mapped on the combined Hg38 and SARS-CoV-2 genome (GenBank: MN908947.3; RefSeq: NC_045512.2) using the BWA-MEM software package with a minimal seed length of 17 (v0.7.17) (*Li and Durbin, 2009*). Only reads mapping uniquely to the SARS-CoV-2 viral genome were retained. Primer sequences were removed after mapping by masking with iVar (*Grubaugh et al., 2019*). From the viral reads BAM (binary alignment map) file, the consensus FASTA file was generated using Samtools (v1.9) (*Li et al., 2009*), mpileup, Bcftools (v 1.9) (*Li et al., 2009*), and SEQTK (https://github.com/lh3/seqtk, *Li, 2021*). For calling low-frequency variants, the viral read alignment file was realigned using the Viterbi method provided by LoFreq (v2.1.2) (*Wilm et al., 2012*). After adding InDel qualities, low-frequency variants were called using LoFreq. Variant filtering was performed with LoFreq and Bcftools (v1.9) (*Li, 2011*). Only variants with a minimum coverage of 75 reads, a minimum phred value of 90, and indels (insertions and deletions) with an HRUN (homopolymer length on the 3' of the variant) below 4 were considered. Based on control experiments described earlier (*Agerer et al., 2021*; *Popa et al., 2020*), all analyses were performed on variants with a minimum alternative frequency of 0.02. Annotations of the variants were performed with SnpEff (v4.3) (*Cingolani et al., 2012b*) and SnpSift (v4.3) (*Cingolani et al., 2012a*). Sequencing data of each sequenced passage were deposited at the European Nucleotide Archive (ENA; https://www.ebi.ac.uk/ena) under project accession PRJEB46926.

## Flow cytometry

For surface staining, single-cell suspensions were treated with TruStain fcX (anti-mouse CD16/32; BioLegend) and Fixable Viability Dye eFluor 780 (eBioscience) according to the manufacturer's instructions. Subsequently, fluorescently labeled antibodies were added to cells and incubated for 20 min at 4°C. Cells were then washed and fixed for 30 min at room temperature using the Fixation Medium of the Fix and Perm Cell Fixation and Permeabilization Kit (Nordic-MUbio). Stained and fixed cell suspensions were analyzed using an LSRFortessa (BD Biosciences). A flow cytometry gating strategy example (infected mouse lung) can be found in *Figure 4—figure supplement 1*. After gating for single/live/CD45$^+$ cells (leukocytes), the following marker combinations have been used to identify the following cell types: lung: alveolar macrophages (AMs): CD11c$^+$/MERTK$^+$; neutrophils: non-AMs/CD11b$^+$/Ly6g$^+$; monocytes (Ly6c$^+$ and Ly6c$^-$): non-AMs/non-neutrophils/CD11b$^+$/CD115$^+$; NK cells: Ly6g$^-$ or MERTK$^-$/NK1.1$^+$/CD19$^-$; B cells: Ly6g$^-$ or MERTK$^-$/CD19$^+$/NK1.1$^-$; T helper cells: Ly6g$^-$ or MERTK$^-$/CD19$^-$/NK1.1$^-$/CD3$^+$ or MHCII$^+$/CD4$^+$/CD8$^-$; cytotoxic T cells: Ly6g$^-$ or MERTK$^-$/CD19$^-$/NK1.1$^-$/CD3$^+$ or MHCII$^+$/CD4$^+$/CD8$^-$; pDCs: Ly6g$^-$ or MERTK$^-$/CD19$^-$/NK1.1$^-$/CD3$^+$ or MHCII$^+$/CD4$^-$/CD8$^-$/CD11c$^+$/BST2$^+$; cDCs: Ly6g$^-$ or MERTK$^-$/CD19$^-$/NK1.1$^-$/CD3$^+$ or MHCII$^+$/CD4$^-$/CD8$^-$/BST2$^-$/CD11c$^+$; blood: neutrophils: CD3$^-$ or CD19$^-$/CD11b$^+$/Ly6g$^+$; monocytes (Ly6c$^+$ and Ly6c$^-$): CD3$^-$ or CD19$^-$/Ly6g$^-$/CD115$^+$/CD11b$^+$; B cells: CD19$^+$/CD3$^-$; T helper cells: CD19$^-$/CD3$^+$/NK1.1$^-$/CD4$^+$/CD8$^-$; cytotoxic T cells: CD19$^-$/CD3$^+$/NK1.1$^-$/CD8$^+$/CD4$^-$; NK cells: CD19$^-$/CD3$^-$/NK1.1$^+$; pDCs: CD19$^-$/CD3$^-$/NK1.1$^-$/CD11c$^+$/BST2$^+$;.

Data were analyzed using FlowJo software (FlowJo LLC) version 10.7. The fluorescently labeled anti-mouse antibodies (all obtained from BioLegend unless specifically indicated) listed in the Key resources table were used throughout this study.

## Plasma cytokine analysis

Plasma cytokine levels were analyzed using the LEGENDplex MU Macrophage/Microglia and Th Cytokine (V02) panels (BioLegend) according to the manufacturer's instructions and using an LSRFortessa. Before flow cytometry analysis, samples were fixed using Fixation Medium of the Fix and Perm Cell Fixation and Permeabilization Kit (Nordic-MUbio).

## Analysis of SARS-CoV-2 Spike protein-specific antibodies

SARS-CoV-2 Spike-specific antibodies were detected by ELISA based on modified, previously described methodologies (*Starkl et al., 2020*). Briefly, Nunc MaxiSorp flat-bottom plates (Thermo

Fisher) were coated overnight at 4°C with 50 µl of purified recombinant SARS-CoV-2 Spike protein ectodomain (**Klausberger et al., 2021**) (produced in CHO-K1 cells), 2 µg/ml in PBS. After washing 3× with PBS 0.05% Tween-20 (Sigma), plates were blocked by incubation with 100 µl of PBS 1% BSA for at least 2 hr at room temperature. Next, plates were washed 3×, followed by incubation with 50 µl of plasma diluted 1:50, 1:200, and 1:800 (for analysis of IgGs) or 1:40 (for analysis of IgA) for 2 hr at 37°C. After another three washing steps, 50 µl biotinylated detection antibodies specific of mouse IgG1 (clone A85-1; BD Pharmingen), IgG2b (clone R12-3; BD Pharmingen), or IgA (polyclonal; Southern Biotech; all diluted 1:1000 in PBS 1% BSA) were added, followed by incubation for 1 hr at room temperature. Plates were then washed again 3× and incubated with 50 µl of horseradish peroxidase-conjugated streptavidin (BD Pharmingen; 1:500 in PBS 1% BSA) for 20 min at room temperature. Finally, plates were washed 5× and detection was performed using the supersensitive TMB liquid substrate (Sigma) and measurement of (after stopping the reaction by addition of 50 µl 2N $H_2SO_4$) absorbance at 450 nm (620 nm reference) on a Sunrise microplate reader (Tecan). IgG titers were calculated by plotting the plasma dilution that gave half-maximal signal of a reference plasma (a plasma pool obtained from mice 14 days after infection with maVie16).

## Histopathological analysis

Lung samples were fixed in 7.5% formalin, embedded in paraffin, cut into 5-µm-thick sections, followed by staining with hematoxylin and eosin or Toluidine Blue. For staining of viral nucleocapsid protein, embedded samples were deparaffinized by immersion in xylene and rehydrated in graded ethanol. After blocking of endogenous peroxidase with 3% $H_2O_2$ in PBS for 10 min at room temperature, tissue sections were first subjected to antigen retrieval using Antigen Unmasking Solution (Vector Labs) for 10 min and then incubated with TRIS-buffered saline containing 0.01% Tween (TBST) and 5% goat serum for 10 min at room temperature. Afterwards, slides were incubated overnight at 4°C with SARS Nucleocapsid Protein Antibody (Novus Biologicals) 1:1000 in TBST 5% goat serum. Subsequently, histological slides were washed with TBST and incubated for 30 min with biotinylated goat anti-rabbit IgG (Vector Labs) 1:200 in TBST 5% goat serum at room temperature. After washing with TBST, tissue sections were processed using the Vectastain ABC kit (Vector Labs) and DAB Substrate kit (Vector Labs) according to the manufacturer's instructions. Tissue sections were finally stained with hematoxylin solution (Mayer's, Sigma-Aldrich), dehydrated, and coverslipped. Histopathological scoring was performed in a blinded fashion by a trained pathologist using the following parameters: alveolar collapse, intra-alveolar exudate, alveolar septal thickening, pneumocyte proliferation, bronchiolar epithelial alteration, bronchitis, peribronchiolar inflammation, interstitial and perivascular infiltrate, and interstitial fibrosis (0 = none, 1 = mild/focal/few, 2 = severe/diffuse).

## Protein modeling

A comparative model of mACE2 was created using the Swiss Modeller (**Waterhouse et al., 2018**), based on the cryoEM structure of hACE2 in complex with BOAT1 (PDB entry 6m18) (**Yan et al., 2020**). Sequence identity was at 82%, and a model with an overall QMEAN score of –1.28 was obtained. The model superimposed to a previous model of the trimeric Spike with hACE2 (https://covid.molssi.org//models/#spike-protein-in-complex-with-human-ace2-ace2-spike-binding) to obtain an initial model of the Spike mACE2 complex (**Capraz et al., 2021**). All proteins were fully glycosylated following our previously described protocols (**Turupcu and Oostenbrink, 2017**). Visualization of the maVie16 mutations in Spike was subsequently created using the mutagenesis wizard in PyMOL (**Schrodinger, 2015**).

## In vivo treatment with recombinant mACE2

rms ACE2 (amino acids 1–740) was produced in CHO cells and purified as described (**Monteil et al., 2020**). To test in vivo ACE2 interference, mice received daily intranasal treatments with 100 µg rmsACE2 (APEIRON Biologics) or the respective dilution of vehicle. Upon prophylactic treatment, the first dose was given as a mix with maVie16 in 50 µl. All remaining doses were administered in 40 µl endotoxin-free PBS (Gibco) to isoflurane anesthetized animals. Intranasal administration to anesthetized mice is an established method that allows administration of compounds to the lower airways via inhalation due to the anesthesia-related low breathing rate and, hence, deep breaths.

## In vivo cytokine depletion

For in vivo depletion of IFNγ and TNF, mice were intraperitoneally injected on days 1 and 3 after maVie16 infection with 200 µl endotoxin-free PBS (Gibco) containing either 500 µg rat anti-mouse IFNγ (clone XMG1.2) and 500 µg rat anti-mouse TNF (clone XT22) or 500 µg rat IgG1 isotype control (clone GL113; specific for ß-galactosidase) obtained from Polpharma Biologics.

## Statistical analysis

Sample size estimation was mainly based on our experience from influenza infection experiments and was performed with R 3.5.1 using relative body weight changes as the main parameter. *t*-test power was calculated using the power.t.test function ('stats' R package). Experimental replicates are always biological replicates. Technical replicates were performed for qPCR data and ELISA data, which are measured in quadruplicates. Statistical analysis was performed using GraphPad Prism 9.1 (GraphPad Software). Details regarding statistical analyses of experiments can be found in the respective figure legends. p-Values ≤ 0.05 were considered statistically significant.

## Acknowledgements

We thank the animal care takers and veterinarians at the animal facility of the Medical University of Vienna for expert help. Dr. Reinhard Grabherr, University of Natural Resources and Life Sciences, Vienna, Austria, kindly provided us with recombinant Spike protein. We thank Hans-Christian Theussl at the Institute for Molecular Biotechnology, Vienna, Austria, for support. We appreciate support by the flow cytometry core facility of the Medical University of Vienna and the Biomedical Sequencing Facility (BSF) jointly run by the Medical University of Vienna and CeMM. SK was supported by the Austrian Science Fund FWF within the special research programs Immunothrombosis (F54-10) and Chromatin Landscapes (F61-04). RG received funding by the Austrian Science Fund FWF (ZK57-B28), and PS acknowledges funding by the Austrian Science Fund (FWF) (P31113-B30). AOR and HS were supported by the FWF (P 34253-B). BA was supported by the Austrian Science Fund (FWF) DK W1212. JMP and the research leading to these results has received funding from the T von Zastrow foundation, the FWF Wittgenstein award (Z271-B19), the Austrian Academy of Sciences, the Innovative Medicines Initiative 2 Joint Undertaking (JU) under grant agreement no. 101005026, and the Canada 150 Research Chairs Program F18-01336 as well as the Canadian Institutes of Health Research COVID-19 grants F20-02343 and F20-02015.

## Additional information

### Competing interests

Gerald Wirnsberger: is an employee of Apeiron Biologics. Apeiron holds a patent on the use of ACE2 for the treatment of lung, heart, or kidney injury and is currently testing soluble ACE2 for treatment in COVID-19 patients. Josef M Penninger: declares a conflict of interest as a founder and shareholder of Apeiron Biologics. Apeiron holds a patent on the use of ACE2 for the treatment of lung, heart, or kidney injury and is currently testing soluble ACE2 for treatment in COVID-19 patients.(patent #WO2021191436A1). The other authors declare that no competing interests exist.

### Funding

| Funder | Grant reference number | Author |
| --- | --- | --- |
| Austrian Science Fund | F54-10 and F61-04 | Sylvia Knapp |
| Austrian Science Fund | ZK57-B28 | Riem Gawish |
| Austrian Science Fund | P31113-B30 | Philipp Starkl |
| Austrian Science Fund | P 34253-B | Anna Ohradanova-Repic Hannes Stockinger |
| Austrian Science Fund | DK W1212 | Benedikt Agerer |

| Funder | Grant reference number | Author |
| --- | --- | --- |
| Austrian Science Fund | Z 271-B19 | Josef M Penninger |
| Canada Research Chairs | F18-01336 | Josef M Penninger |
| Canadian Institutes of Health Research | F20-02343 and F20-02015 | Josef M Penninger |
| Innovative Medicines Initiative | 101005026 | Josef M Penninger |

The funders had no role in study design, data collection and interpretation, or the decision to submit the work for publication.

## Author contributions

Riem Gawish, Philipp Starkl, Conceptualization, Data curation, Formal analysis, Investigation, Methodology, Writing – original draft, Writing – review and editing; Lisabeth Pimenov, Anna Ohradanova-Repic, Formal analysis, Investigation; Anastasiya Hladik, Karin Lakovits, Shane JF Cronin, Astrid Hagelkruys, Nuria Montserrat, Investigation; Felicitas Oberndorfer, Tümay Capraz, Jan W Perthold, Formal analysis; Gerald Wirnsberger, Investigation, Supervision; Benedikt Agerer, Lukas Endler, Data curation, Formal analysis; Domagoj Cikes, Rubina Koglgruber, Methodology, Resources; Ali Mirazimi, Louis Boon, Hannes Stockinger, Andreas Bergthaler, Supervision; Chris Oostenbrink, Formal analysis, Supervision, Visualization, Writing – original draft; Josef M Penninger, Supervision, Writing – review and editing; Sylvia Knapp, Conceptualization, Funding acquisition, Investigation, Supervision, Writing – original draft, Writing – review and editing

## Author ORCIDs

Riem Gawish ![ORCID] http://orcid.org/0000-0003-4267-2131
Philipp Starkl ![ORCID] http://orcid.org/0000-0001-7521-129X
Anna Ohradanova-Repic ![ORCID] http://orcid.org/0000-0002-8005-8522
Hannes Stockinger ![ORCID] http://orcid.org/0000-0001-6404-4430
Chris Oostenbrink ![ORCID] http://orcid.org/0000-0002-4232-2556
Sylvia Knapp ![ORCID] http://orcid.org/0000-0001-9016-5244

## Ethics

All experiments involving SARS-CoV-2 or its derivatives were performed in Biosafety Level 3 (BSL-3) facilities at the Medical University of Vienna and performed according to the ethical guidelines and after approval by the institutional review board of the Austrian Ministry of Sciences (BMBWF-2020-0.253.770) and in accordance with the directives of the EU.

## Decision letter and Author response

Decision letter https://doi.org/10.7554/eLife.74623.sa1
Author response https://doi.org/10.7554/eLife.74623.sa2

# Additional files

## Supplementary files

• Transparent reporting form

## Data availability

maVie16 SARS-CoV-2 genome sequence will be published on: https://www.ebi.ac.uk/ena Project accession: PRJEB46926.

The following dataset was generated:

| Author(s) | Year | Dataset title | Dataset URL | Database and Identifier |
| --- | --- | --- | --- | --- |
| CeMM Research Center for Molecular Medicine of the Austrian Academy of Sciences | 2021 | Exploring COVID-19 disease, immunity and therapeutic options in mice using maVie16, a host-adapted SARS-CoV-2 | https://www.ebi.ac.uk/ena/browser/view/PRJEB46926 | European Nucleotide Archive, PRJEB46926 |

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
