## [Editor Report]

To establish a mouse model for the SARS-CoV-2 infection, Gawish and colleagues performed serial passage of a human virus isolate in mice. They show that the mouse-adapted SARS-CoV-2 variant remains dependent on ACE2 for efficient infection, recapitulates some clinical characteristics of COVID-19, and acquired some changes also found in the Omicron variant of concern. Finally, the authors demonstrate that inhalation of recombinant ACE2 protected mice from mouse COVID-19, suggesting that this model will be useful for the testing of antiviral agents.

---

## [Decision Letter]

**Decision letter after peer review:**

Thank you for submitting your article "ACE2 is the critical in vivo receptor for SARS-CoV-2 in a novel COVID-19 mouse model with TNF- and IFNγ-driven immunopathology" for consideration by *eLife*. Your article has been reviewed by 2 peer reviewers, including Frank Kirchhoff as Reviewing Editor and Reviewer #1, and the evaluation has been overseen by a Reviewing Editor and Tadatsugu Taniguchi as the Senior Editor.

Essential revisions:

1. Infection is also dependent on proteolytic activation of the Spike protein by TMPRSS2 or Cathepsins. Please clarify whether any of the mutations in Spike affect potential cleavage sites and how conserved these proteases are between men and mice?

2. Mice show a drop in body temperature while humans develop fever. This difference should be explained for the non-expert. Similarly, the reasons for severe weight loss (loss of appetite, diarrhoea, other?) should briefly be mentioned.

3. The authors emphasize the similarities to human disease. They should also discuss differences; e.g. it seems that the incubation time is much shorter.

4. As mentioned by the authors, ACE2 is essential for infection and viral replication of mouse-adapted SARS-CoV-2 in mice. It seems more plausible to just show that ACE-KO mice are resistant to infection. If that is the case, it seems self-evident that they do not develop disease. Either a better rationale for studying clinical parameters in KO mice in the absence of infection should be provided or this part should be shortened substantially.

5. IFNs drive inflammation but also induce antiviral factors. Please clarify whether the induction of type I and II IFN responses associated with declines in viral loads?

6. It is stated that recombinant ACE2 was inhaled. This suggests that the protein was aerosolized and applied as spray. Add the respective description in the Materials and methods section.

7. "albeit 1 x 105 (and higher) TCID50 induced a profound but transient body weight loss of about 15-20% (Figure 3D to F)." Only panel D shows body weight and the loss loss seems generally <20%. Clarify and revise.

*Reviewer #1 (Recommendations for the authors):*

Infection is also dependent on proteolytic activation of the Spike protein by TMPRSS2 or Cathepsins. Did any of the mutations in Spike affect potential cleavage sites and how conserved are these proteases between men and mice?

Mice show a drop in body temperature while humans develop fever. This difference should be explained for the non-expert. Similarly, the reasons for severe weight loss (loss of appetite, diarrhoea, other?) should briefly be mentioned.

The authors emphasize the similarities to human disease. They should also discuss differences; e.g. it seems that the incubation time is much shorter.

As mentioned by the authors, ACE2 is essential for infection and viral replication of mouse-adapted SARS-CoV-2 in mice. It seems more plausible to just show that ACE-KO mice are resistant to infection. If that is the case, it seems self-evident that they do not develop disease. Either a better rationale for studying clinical parameters in KO mice in the absence of infection should be provided or this part should be shortened substantially.

IFNs drive inflammation but also induce antiviral factors. Was the induction of type I and II IFN responses associated with declines in viral loads?

---

## [Author Response]

Essential revisions:1. Infection is also dependent on proteolytic activation of the Spike protein by TMPRSS2 or Cathepsins. Please clarify whether any of the mutations in Spike affect potential cleavage sites and how conserved these proteases are between men and mice?

We thank the reviewer for highlighting this interesting and important aspect. Proteolytic processing of SARS-CoV-2 Spike at the S1/S2 (681-PRRAR; cleaved by furin) and the S2' (815-R; cleaved by TMPRSS2) cleavage sites is indeed important for infection. In the absence of furin and TMPRSS2, (less efficient) cathepsin L Spike processing at both sites has been suggested (Takeda, Microbiology and Immunology, 2021; PMID 34561887).

Of note, both cleavage sites are intact in maVie16.

Johnson, Xie et al., recently reported (Nature, 2021; PMID 33494095) that SARS-CoV-2 with a mutated S1/S2 cleavage site shows markedly decreased replication and pathogenicity in K18hACE2 mice (expressing human ACE2 under control of the K18 promoter), indicating that proteolytic (furin) cleavage is also an important factor in mouse models of COVID-19. This suggests that mice have the relevant proteases required for proteolytic SARS-CoV-2 processing. While the amino acid sequence similarity of furin is more than 90% between mouse and man, that of TMPRSS2 is approx. 80% (Vaarala et al., J. Pathol, 2001; PMID 11169526). Together we believe that the mutations in maVie16s Spike receptor binding domain (RBD) do not interfere with proteolytic processing and that original resistance of mice to infection with the human SARS-CoV-2 isolate Bav-Pat-1 is not due to differences in murine protease expression, but rather because the human virus cannot efficiently bind to mouse ACE2. We have now added the following paragraphs to our manuscript to address this issue:

“Since the observed Spike mutations are distant from putative cleavage sites (Takeda, 2021), it is likely that mACE2 affinity is increased without affecting proteolytic processing by proteases such as furin or TMPRSS2 (Takeda, 2021).” (page 6)

“At the same time, the multibasic S1/S2 furin cleavage site (681-PRRAR) as well as the S2’ cleavage site (at 815-R) were maintained, rendering maVie16 Spike accessible for proteolytic processing by endogenous proteases such as TMPRSS2 or furin, which are essential for productive infection (Takeda, 2021) and highly conserved between mice and men (Vaarala et al., 2001).” (page12)

2. Mice show a drop in body temperature while humans develop fever. This difference should be explained for the non-expert. Similarly, the reasons for severe weight loss (loss of appetite, diarrhoea, other?) should briefly be mentioned.

Thanks for bringing up these important points. We added a short comment to the Results section for clarification:

“Mice infected with passage 15 furthermore exhibited a severe drop in body temperature (Figure 1D), which is the regulatory response of mice to severe inflammation at room temperature (Garami et al., 2018).” (page 5)

While weight loss is a (the) major readout for systemic pathology in mouse models of viral infections, we can only speculate about the reason for the drop in bodyweight upon maVie16 infection. Previous studies revealed reduced food intake (anorexia) as the main reason for weight loss upon influenza infection in mice, whereas weight loss in other virus infections (e.g. lymphocytic choriomeningitis virus) was caused by a combination of anorexia and cachexia (Baazim et al., Nat Rev Immunology, 2021; PMID: 34608281). Notably, as we did not observe diarrhoea in infected mice, this can be excluded as potential explanation.

To address this issue in our manuscript, we added the following sentence (and a reference) to the Results section:

“a progressive loss of body weight of mice infected with later stage passaged SARS-CoV-2 indicated enhanced pathogenicity of the virus (Figure 1C). Weight loss is a sign of disease severity in many rodent models of infection, and related to anorexia and/or cachexia (Baazim et al., 2021).” (page 5)

3. The authors emphasize the similarities to human disease. They should also discuss differences; e.g. it seems that the incubation time is much shorter.

We thank the reviewer for this critical comment. While in humans, the typical incubation time between infection and symptom onset is around 4-5 days (Li, Guan, Wu, Wang, et al., NEJM, 2020; PMID: 31995857), maVie16-infected mice develop clinical signs of disease around day 2 post infection. One explanation for this difference might be the route of infection and the exposure dose. In mice, intranasal administration of maVie16 results in deep inhalation, and immediate exposure of the lower airways to virus. In contrast, humans usually get infected via droplets that first reach the upper airways (nasopharynx), where the virus replicates to then “move” down to the lower airways.

We have introduced a paragraph to highlight this difference in the discussion:

“The time to symptom onset was approximately 2-3 days, which is faster than reported in humans (around 5 days) (Li et al., 2020), and might be explained by the route of infection and subsequent high viral dose in the lower airways, as opposed to droplet infection of the upper respiratory tract in humans.” (page 12)

4. As mentioned by the authors, ACE2 is essential for infection and viral replication of mouse-adapted SARS-CoV-2 in mice. It seems more plausible to just show that ACE-KO mice are resistant to infection. If that is the case, it seems self-evident that they do not develop disease. Either a better rationale for studying clinical parameters in KO mice in the absence of infection should be provided or this part should be shortened substantially.

We fully agree that ACE2 deficiency likely renders mice resistant to infection and did not intend to speculate about other possibilities. Resistance to infection is also supported by our data showing a complete absence of nucleocapsid positive cells (Figure 7E) in lungs of infected Ace2^-/y^ animals. We apologize for the confusion and have rephrased and shortened this part of our manuscript:

“In contrast to infected littermate control animals, Ace2^-/y^ mice were fully protected against infection with 5 x 10^5^ TCID50 maVie16, maintained stable body weight (Figure 7B) and temperature (Figure 7C) and were protected from pneumonia development as indicated by lower lung weight (Figure 7D) and the absence of any lung pathology (Figure 7E). Resistance to infection was further indicated by the complete absence of nucleocapsid-positive cells in lungs of Ace2^-/y^ animals (Figure 7E). In conflict with this observation, we still detected similar viral genome copy numbers in both groups (Figure 7—figure supplement 1B), suggesting that this was reflective of initial maVie16 input rather than productive infection.” (page 10)

5. IFNs drive inflammation but also induce antiviral factors. Please clarify whether the induction of type I and II IFN responses associated with declines in viral loads?

We show that maVie16 induces a fast and robust plasmacytoid dendritic cell (pDC) response (Figure 4C), an important source of type I IFN, as well as expression of PKR (Figure 4D; eif2ak2), a typical antiviral factor induced by type I IFN, as early as 2 days post infection. This correlates with the peak of viral burden, which progressively declines from this point (Figure 5B). However, as others have shown that IFNs play an ambiguous role during COVID-19 and our study still lacks experiments with IFNAR1 deficient animals or type I IFN blocking antibodies, a causality between type I IFN and viral clearance in our modal cannot be claimed. In contrast to type I IFN, lung IFNγ levels peak on day 7 post infection, when lung pathology is most severe but maVie16 is already cleared (based on absence of nucleocapsid-positive lung cell). We conclude from our experiments using IFNγ blocking antibodies (Figure 6D-G and Figure 6—figure supplement 2) that type II IFN (in combination with TNF) rather contributes to immunopathology than to viral clearance in our model.

Immunopathologic features of type II IFN are already addressed in the Discussion section of our manuscript (page 12-13).

In addition, we have now added the following paragraph to further explain the role of type I IFN in our model:

“In addition, mCOVID-19 is characterized by an early pDC mobilization in blood and lung, associated with a fast pulmonary type I IFN response. This correlates with the peak of viral burden, which subsequently declines. However, as others have shown that IFNs play an ambiguous role during COVID-19 (Israelow et al., 2020), further experiments (e.g. using IFNAR1-deficient animals or type I IFN blocking antibodies) will be required to address the direct impact of type I IFNs on maVie16 clearance in vivo.” (page 12)

6. It is stated that recombinant ACE2 was inhaled. This suggests that the protein was aerosolized and applied as spray. Add the respective description in the Materials and methods section.

Thank you for this comment. We administered recombinant ACE2 via intranasal inhalation of anesthetized mice (which reduces the respiratory rate and ensures deep breaths, i.e. comparable to inhalation using asthma sprays), an established method to ensure application of compounds to the lower airways. To clarify this aspect, we added the following explanation to the methods section:

“Intranasal administration to anesthetized mice is an established method that allows administration of compounds to the lower airways via inhalation, due to the anesthesia-related low breathing rate and, hence, deep breaths” (page 25)

7. "albeit 1 x 105 (and higher) TCID50 induced a profound but transient body weight loss of about 15-20% (Figure 3D to F)." Only panel D shows body weight and the loss loss seems generally <20%. Clarify and revise.

Thanks for bringing this to our attention. We originally decided to state 15-20% because as individual mice exhibited a weight loss decrease up to 19%. However, we realize that this is not clearly recognizable from the displayed mean value. To avoid confusion, we changed this statement to “15%” (page 7).

Reviewer #1 (Recommendations for the authors):Infection is also dependent on proteolytic activation of the Spike protein by TMPRSS2 or Cathepsins. Did any of the mutations in Spike affect potential cleavage sites and how conserved are these proteases between men and mice?

We thank the reviewer for highlighting this interesting and important aspect. Proteolytic processing of SARS-CoV-2 Spike at the S1/S2 (681-PRRAR; cleaved by furin) and the S2' (815-R; cleaved by TMPRSS2) cleavage sites is indeed important for infection. In the absence of furin and TMPRSS2, (less efficient) cathepsin L Spike processing at both sites has been suggested (Takeda, Microbiology and Immunology, 2021; PMID 34561887).

Of note, both cleavage sites are intact in maVie16.

Johnson, Xie et al., recently reported (Nature, 2021; PMID 33494095) that SARS-CoV-2 with a mutated S1/S2 cleavage site shows markedly decreased replication and pathogenicity in K18hACE2 mice (expressing human ACE2 under control of the K18 promoter), indicating that proteolytic (furin) cleavage is also an important factor in mouse models of COVID-19. This suggests that mice have the relevant proteases required for proteolytic SARS-CoV-2 processing. While the amino acid sequence similarity of furin is more than 90% between mouse and man, that of TMPRSS2 is approx. 80% (Vaarala et al., J. Pathol, 2001; PMID 11169526). Together we believe that the mutations in maVie16s Spike receptor binding domain (RBD) do not interfere with proteolytic processing and that original resistance of mice to infection with the human SARS-CoV-2 isolate Bav-Pat-1 is not due to differences in murine protease expression, but rather because the human virus cannot efficiently bind to mouse ACE2. We have now added the following paragraphs to our manuscript to address this issue:

“Since the observed Spike mutations are distant from putative cleavage sites (Takeda, 2021), it is likely that mACE2 affinity is increased without affecting proteolytic processing by proteases such as furin or TMPRSS2 (Takeda, 2021).” (page 6)

“At the same time, the multibasic S1/S2 furin cleavage site (681-PRRAR) as well as the S2’ cleavage site (at 815-R) were maintained, rendering maVie16 Spike accessible for proteolytic processing by endogenous proteases such as TMPRSS2 or furin, which are essential for productive infection (Takeda, 2021) and highly conserved between mice and men (Vaarala et al., 2001).” (page12)

Mice show a drop in body temperature while humans develop fever. This difference should be explained for the non-expert. Similarly, the reasons for severe weight loss (loss of appetite, diarrhoea, other?) should briefly be mentioned.

Thanks for bringing up these important points. We added a short comment to the Results section for clarification: „Mice infected with passage 15 furthermore exhibited a severe drop in body temperature (Figure 1D), which is the regulatory response of mice to severe inflammation at room temperature (Garami et al., 2018).” (page 5)

While weight loss is a (the) major readout for systemic pathology in mouse models of viral infections, we can only speculate about the reason for the drop in bodyweight upon maVie16 infection. Previous studies revealed reduced food intake (anorexia) as the main reason for weight loss upon influenza infection in mice, whereas weight loss in other virus infections (e.g. lymphocytic choriomeningitis virus) was caused by a combination of anorexia and cachexia (Baazim et al., Nat Rev Immunology, 2021; PMID: 34608281). Notably, as we did not observe diarrhea in infected mice, this can be excluded as potential explanation.

To address this issue in our manuscript, we added the following sentence (and a reference) to the Results section:

“a progressive loss of body weight of mice infected with later stage passaged SARS-CoV-2 indicated enhanced pathogenicity of the virus (Figure 1C). Weight loss is a sign of disease severity in many rodent models of infection, and related to anorexia and/or cachexia (Baazim et al., 2021).” (page 5)

The authors emphasize the similarities to human disease. They should also discuss differences; e.g. it seems that the incubation time is much shorter.

We thank the reviewer for this critical comment. While in humans, the typical incubation time between infection and symptom onset is around 4-5 days (Li, Guan, Wu, Wang, et al., NEJM, 2020; PMID: 31995857), maVie16-infected mice develop clinical signs of disease around day 2 post infection. One explanation for this difference might be the route of infection and the exposure dose. In mice, intranasal administration of maVie16 results in deep inhalation, and immediate exposure of the lower airways to virus. In contrast, humans usually get infected via droplets that first reach the upper airways (nasopharynx), where the virus replicates to then “move” down to the lower airways.

We have introduced a paragraph to highlight this difference in the discussion:

“The time to symptom onset was approximately 2-3 days, which is faster than reported in humans (around 5 days) (Li et al., 2020), and might be explained by the route of infection and subsequent high viral dose in the lower airways, as opposed to droplet infection of the upper respiratory tract in humans.” (page 12)

As mentioned by the authors, ACE2 is essential for infection and viral replication of mouse-adapted SARS-CoV-2 in mice. It seems more plausible to just show that ACE-KO mice are resistant to infection. If that is the case, it seems self-evident that they do not develop disease. Either a better rationale for studying clinical parameters in KO mice in the absence of infection should be provided or this part should be shortened substantially.

We fully agree that ACE2 deficiency likely renders mice resistant to infection and did not intend to speculate about other possibilities. Resistance to infection is also supported by our data showing a complete absence of nucleocapsid positive cells (Figure 7E) in lungs of infected Ace2^-/y^ animals. We apologize for the confusion and have rephrased and shortened this part of our manuscript:

“In contrast to infected littermate control animals, Ace2^-/y^ mice were fully protected against infection with 5 x 10^5^ TCID50 maVie16, maintained stable body weight (Figure 7B) and temperature (Figure 7C) and were protected from pneumonia development as indicated by lower lung weight (Figure 7D) and the absence of any lung pathology (Figure 7E). Resistance to infection was further indicated by the complete absence of nucleocapsid-positive cells in lungs of Ace2^-/y^ animals (Figure 7E). In conflict with the absence of nucleocapsid-positive cells, we still detected similar viral genome copy numbers in both groups (Figure 7—figure supplement 1B), suggesting that this was reflective of initial maVie16 input rather than productive infection.” (page 10)

IFNs drive inflammation but also induce antiviral factors. Was the induction of type I and II IFN responses associated with declines in viral loads?

We show that maVie16 induces a fast and robust plasmacytoid dendritic cell (pDC) response (Figure 4C), an important source of type I IFN, as well as expression of PKR (Figure 4D; eif2ak2), a typical antiviral factor induced by type I IFN, as early as 2 days post infection. This correlates with the peak of viral burden, which progressively declines from this point (Figure 5B). However, as others have shown that IFNs play an ambiguous role during COVID-19 and our study still lacks experiments with IFNAR1 deficient animals or type I IFN blocking antibodies, a causality between type I IFN and viral clearance in our modal cannot be claimed. In contrast to type I IFN, lung IFNγ levels peak on day 7 post infection, when lung pathology is most severe but maVie16 is already cleared (based on absence of nucleocapsid-positive lung cell). We conclude from our experiments using IFNγ blocking antibodies (Figure 6D-G and Figure 6—figure supplement 2) that type II IFN (in combination with TNF) rather contributes to immunopathology than to viral clearance in our model.

Immunopathologic features of type II IFN are already addressed in the Discussion section of our manuscript (page 12-13).

In addition, we have now added the following paragraph to further explain the role of type I IFN in our model:

“In addition, mCOVID-19 is characterized by an early pDC mobilization in blood and lung, associated with a fast pulmonary type I IFN response. This correlates with the peak of viral burden, which subsequently declines. However, as others have shown that IFNs play an ambiguous role during COVID-19 (Israelow et al., 2020), further experiments (e.g. using IFNAR1-deficient animals or type I IFN blocking antibodies) will be required to address the direct impact of type I IFNs on maVie16 clearance in vivo.” (page 12)

References

Dinnon, K. H., 3rd, Leist, S. R., Schafer, A., Edwards, C. E., Martinez, D. R., Montgomery, S. A., West, A., Yount, B. L., Jr., Hou, Y. J., Adams, L. E., Gully, K. L., Brown, A. J., Huang, E., Bryant, M. D., Choong, I. C., Glenn, J. S., Gralinski, L. E., Sheahan, T. P., and Baric, R. S. (2020). A mouseadapted model of SARS-CoV-2 to test COVID-19 countermeasures. Nature, 586(7830), 560566. https://doi.org/10.1038/s41586-020-2708-8

Huang, K., Zhang, Y., Hui, X., Zhao, Y., Gong, W., Wang, T., Zhang, S., Yang, Y., Deng, F., Zhang, Q., Chen, X., Yang, Y., Sun, X., Chen, H., Tao, Y. J., Zou, Z., and Jin, M. (2021). Q493K and Q498H substitutions in Spike promote adaptation of SARS-CoV-2 in mice. EBioMedicine, 67, 103381. https://doi.org/10.1016/j.ebiom.2021.103381

Leist, S. R., Dinnon, K. H., 3rd, Schafer, A., Tse, L. V., Okuda, K., Hou, Y. J., West, A., Edwards, C. E., Sanders, W., Fritch, E. J., Gully, K. L., Scobey, T., Brown, A. J., Sheahan, T. P., Moorman, N. J., Boucher, R. C., Gralinski, L. E., Montgomery, S. A., and Baric, R. S. (2020). A Mouse-Adapted SARS-CoV-2 Induces Acute Lung Injury and Mortality in Standard Laboratory Mice. Cell, 183(4), 1070-1085 e1012. https://doi.org/10.1016/j.cell.2020.09.050

Takeda, M. (2021). Proteolytic activation of SARS-CoV-2 spike protein. Microbiol Immunol. https://doi.org/10.1111/1348-0421.12945

Tegally, H., Wilkinson, E., Giovanetti, M., Iranzadeh, A., Fonseca, V., Giandhari, J., Doolabh, D., Pillay, S., San, E. J., Msomi, N., Mlisana, K., von Gottberg, A., Walaza, S., Allam, M., Ismail, A., Mohale, T., Glass, A. J., Engelbrecht, S., Van Zyl, G.,. de Oliveira, T. (2020). Emergence and rapid spread of a new severe acute respiratory syndrome-related coronavirus 2 (SARSCoV-2) lineage with multiple spike mutations in South Africa. medRxiv, 2020.2012.2021.20248640. https://doi.org/10.1101/2020.12.21.20248640

Vaarala, M. H., Porvari, K. S., Kellokumpu, S., Kyllonen, A. P., and Vihko, P. T. (2001). Expression of transmembrane serine protease TMPRSS2 in mouse and human tissues. J Pathol, 193(1), 134-140. https://doi.org/10.1002/1096-9896(2000)9999:9999<::AID-PATH743>3.0.CO;2-T

Voloch, C. M., Silva F, R. d., de Almeida, L. G. P., Cardoso, C. C., Brustolini, O. J., Gerber, A. L., Guimarães, A. P. d. C., Mariani, D., Costa, R. M. d., Ferreira, O. C., Cavalcanti, A. C., Frauches, T. S., de Mello, C. M. B., Galliez, R. M., Faffe, D. S., Castiñeiras, T. M. P. P., Tanuri, A., and de Vasconcelos, A. T. R. (2020). Genomic characterization of a novel SARS-CoV-2 lineage from Rio de Janeiro, Brazil. medRxiv, 2020.2012.2023.20248598. https://doi.org/10.1101/2020.12.23.20248598

Wang, J., Shuai, L., Wang, C., Liu, R., He, X., Zhang, X., Sun, Z., Shan, D., Ge, J., Wang, X., Hua, R., Zhong, G., Wen, Z., and Bu, Z. (2020). Mouse-adapted SARS-CoV-2 replicates efficiently in the upper and lower respiratory tract of BALB/c and C57BL/6J mice. Protein Cell, 11(10), 776-782. https://doi.org/10.1007/s13238-020-00767-x